# Controlling periodic long-range signalling to drive a morphogenetic transition

**Hugh Z Ford[1], Angelika Manhart[2,3], Jonathan R Chubb[1]\***

[1]Laboratory for Molecular Cell Biology and Department of Cell and Developmental Biology, University College London, London, United Kingdom; [2]Department of Mathematics, University College London, London, United Kingdom; [3]Faculty of Mathematics, University of Vienna, Vienna, Austria

**Abstract** Cells use signal relay to transmit information across tissue scales. However, the production of information carried by signal relay remains poorly characterised. To determine how the coding features of signal relay are generated, we used the classic system for long-range signalling: the periodic cAMP waves that drive *Dictyostelium* collective migration. Combining imaging and optogenetic perturbation of cell signalling states, we find that migration is triggered by an increase in wave frequency generated at the signalling centre. Wave frequency is regulated by cAMP wave circulation, which organises the long-range signal. To determine the mechanisms modulating wave circulation, we combined mathematical modelling, the general theory of excitable media, and mechanical perturbations to test competing models. Models in which cell density and spatial patterning modulate the wave frequency cannot explain the temporal evolution of signalling waves. Instead, our evidence leads to a model where wave circulation increases the ability for cells to relay the signal, causing further increase in the circulation rate. This positive feedback between cell state and signalling pattern regulates the long-range signal coding that drives morphogenesis.

## Editor's evaluation

This fundamental work substantially advances our understanding of how multicellular structures transmit information over long ranges. Compelling approaches combining experiments and theory unravel the mechanism by which amoeba form migrating cellular waves by chemotaxis. The work will be of broad interest to cell and developmental biologists.

*For correspondence:
j.chubb@ucl.ac.uk

**Competing interest:** The authors declare that no competing interests exist.

## Introduction

Tissues require long-range signalling strategies to coordinate cell behaviour. As signal diffusion is only efficient over a few cell diameters, cells relay signals to rapidly disseminate information over tissue scales (*Gelens et al., 2014*). Signals are relayed by the activation of cells in response to active neighbours, which in turn activate their own neighbours (*Dieterle et al., 2020*). In this way, long-range signalling through tissues is propagated as waves of cell activation that travel outwardly from signalling centres (*Deneke and Di Talia, 2018*). Prominent examples of biological activation waves include ERK waves in the developing skin, breast, and colon and during wound closure (*Aoki et al., 2017*; *Ender et al., 2022*; *Pond et al., 2022*), periodic activation waves in the brain (*Huang et al., 2010*), chemoattractant waves in neutrophil swarms (*Afonso et al., 2012*), morphogen waves during gastrulation (*Liu et al., 2021*), waves of cell stress in bacterial biofilms (*Chou et al., 2022*), and cAMP waves in *Dictyostelium* aggregation (*Tomchik and Devreotes, 1981*). Waves can also be pathological, such as in cardiac fibrillation, where waves drive muscle contraction independently of the pacemaker (*Pandit*

*and Jalife, 2013*). In these examples, the behaviour of the cells is organised by a self-sustaining signalling pattern; however, it is unclear how cell populations regulate these patterns.

One of the most intensively studied paradigms for signal relay is the *Dictyostelium* transition from a single-cell state to a multicellular state (*Weijer, 2004*). Upon starvation, single cells begin to communicate via cAMP (*Tomchik and Devreotes, 1981*; *Gregor et al., 2010*). Upon receipt of cAMP, cells secrete their own cAMP, which then activates the next set of cells, hence propagating the signal. As cAMP is a chemoattractant, cells move towards signalling centres, resulting in aggregation into the multicellular phase of their lifecycle. At the population level, the cAMP signal propagates in waves with circular and spiral patterns characteristic of other relay systems (*Durston, 1973*; *Lee et al., 1996*).

Despite decades of research into cAMP signalling in *Dictyostelium,* the mechanism of information production and control remains poorly characterised. Several lines of evidence suggest information is carried by cAMP wave gradient, frequency, and speed and decoded by cells in terms of changes in motility (*Siegert and Weijer, 1989*). One view is that cells interpret an increase in the frequency of the cAMP waves as a trigger to begin collective chemotaxis (*Skoge et al., 2014*). Alternative views are that other changing features of the waves, such as their speed and gradient, convey information instructing chemotaxis (*Fisher et al., 1989*; *Wessels et al., 1992*; *Song et al., 2006*; *Nakajima et al., 2014*). The current paradigm explaining the changes in wave behaviour over developmental time is that the excitability (the ability to relay the signal) of cell groups increases (individual cell excitability remains constant) at higher cell densities, which then manifests in the changing wave dynamics coupled to the onset of aggregation (*Höfer et al., 1995*; *van Oss et al., 1996*; *Dallon and Othmer, 1997*). One caveat with this view is that it requires some degree of aggregation to initiate the signal that drives the aggregation process. This paradigm was informed by extensive theory and studies on excitable chemical systems. In these systems, changes in wave dynamics emerge from rotating spiral waves whose rotation rate and curvature specifies the frequency and speed of activation waves. The curvature and rotation rate are specified by the circulation of the signal at the signalling centre (*Tyson and Keener, 1988*; *Winfree, 2001*). However, the major limitation in uniting this theory with experiment to decipher wave generation and interpretation is that studies have looked either at macroscopic (population) or microscopic (cell) levels but not both at the same time. As a result, it is not clear how cell signalling, movement, and spatial patterning at the signalling centre modulate the periodic cAMP waves that coordinate this morphogenetic transition.

To determine how population-level signalling dynamics are modulated by the signalling centre, we have leveraged a sensitive cAMP reporter to characterise collective cell signalling and motion during *Dictyostelium* aggregation, with single-cell resolution, over developmental time and length scales. We find that the onset of cell aggregation coincides with an increase in wave frequency and decrease in wave speed. Optogenetic activation of cAMP signalling with defined frequencies reveals how an increase in the frequency of wave production from the signalling centre is required for efficient aggregation. These changing wave dynamics result from the increasing circulation rate of cAMP waves at the signalling centre. We show that the circulation of cAMP waves spatially organises the signal as a spiral pattern, and the rotation rate and curvature of the spiral sets wave frequency and speed. We evaluate contrasting established and new hypotheses by which the circulation dynamics are regulated. We challenge the current paradigm, that increasing cell density drives an increase in collective cell excitability, and show that a model in which wave dynamics result from cell rearrangement at the signalling centre also cannot explain the observed signalling patterns. Instead, we show that the ability for individual cells to relay signalling waves increases as the wave frequency increases. This creates positive feedback that gradually increases the rate of wave circulation and the ability for cells to relay the signal. Together these observations imply the dynamic behaviour of signalling driving morphogenesis results from a mutual reinforcement between cell state and signal pattern.

## Results

### The onset of collective cell signalling

To track the signalling state of individual cells during morphogenesis, we imaged cells expressing the intensiometric cAMP signalling reporter Flamindo2 (*Odaka et al., 2014*; *Hashimura et al., 2019*) at an endogenous genetic locus. For simultaneous analysis of cell motion, the cells also expressed a nuclear marker (H2B-mCherry) for cell tracking (*Corrigan and Chubb, 2014*). The fluorescence of

Flamindo2 has an inverse relationship to the intracellular [cAMP]. This allows visualisation of the onset and progression of cAMP signalling wave patterns over cell populations during early development (*Figure 1A* and *Figure 1—video 1*). To monitor the motion and signalling of individual cells in their population context, we tracked the central position of the nuclei and measured the internal [cAMP] from the mean Flamindo2 intensity of the cell body (*Figure 1B*, *Figure 1—figure supplement 1A* and *Figure 1—video 2*). Analysing all cell tracks simultaneously provided the means to study the onset and adaptation of signalling behaviour and collective motion of single cells in their population context (*Figure 1C–E*).

Early during development, cells predominantly fired asynchronously. The proportion of spontaneously activating cells declined as the signalling behaviour became more correlated, with periodic waves of signalling including an increasing proportion of the cells (*Figure 1D*). Prior to the onset of cAMP waves (around 4 hr after starvation), cells produced cAMP pulses spontaneously at a mean rate of around 1 pulse per 12 min, with a minimum of around 1 pulse per 4 min (*Figure 1E*). During this time, the population distribution of inter-activation intervals follows a gamma distribution (*Figure 1E* and *Figure 1—figure supplement 1B*). We interpret this fit as there being a minimum waiting time between pulses (a refractory phase), with the exponential tail of the distribution suggesting the probability of firing is independent of the previous firing event.

The initial cAMP waves (around 4–5 hr) are broad bands of broken waves amalgamating from numerous circular patterns that expand from spontaneously activating cells (*Figure 1A*). Activation events prior to synchronisation are spontaneous, spatially variable, and mostly do not initiate a cAMP wave. Together with the independence of successive firing events, these observations are inconsistent with the model that individual 'pacemaker' cells act as the organising source of global cAMP waves. However, it may be possible that cell clusters (arising for example from random motion) may collectively act as a pacemaker (*Mirollo and Strogatz, 1990*; *Gregor et al., 2010*). For example, if the cell cluster is sufficiently dense such that at least one cell in the cluster is likely to activate just after the refractory time. Nevertheless, any given cell will experience circular waves from several different sources with a variable periodicity (5–9 min).

Circular waves produce cAMP signals at irregular intervals which are typically greater than the inter-pules interval (*Figure 1D*). Consequently, not all cells will relay a circular wave because cells that fire synchronously with one wave can potentially reactivate and so be refractory for the next wave. This is highlighted by a multi-peaked population distribution for inter-activation intervals where the smallest peak is associated with spontaneous cAMP pulses, while the other peaks are associated with synchronised firing during a wave (*Figure 1E* and *Figure 1—figure supplement 1B*). Rotating spiral cAMP waves form from broken waves around 5–6 hr following starvation (*Figure 1A*). To interrogate how waves break and give rise to spiral waves, we engineered cells to express blue-photoactivable adenylyl cyclase (bPAC; *Stierl et al., 2011*) to control the time and place of cAMP waves by exposing a subset of cells to blue or green light (*Figure 1—figure supplement 2* and *Figure 1—video 3*). In the absence of light pulses, we observed a steady train of periodic waves (*Figure 1—figure supplement 2B*). Photoactivation of a small patch of cells generated a circular wave which fused with oncoming waves (*Figure 1—figure supplement 2A* first and second row and *Figure 1—figure supplement 2C*). Wave breaks (*Figure 1—figure supplement 2A* forth – sixth row) occurred only when the oncoming wave collided with an asymmetric wave generated by the activated patch (*Figure 1—figure supplement 2A* third row). We interpret the asymmetry arising when cells activate at the border of two regions; one where cells are in a refractory state (preventing wave propagation) and the other in which cells are recovered (allowing wave propagation; *Figure 1—figure supplement 2D*). This experiment suggests that spiral waves might naturally arise from spontaneous cell activation behind a circular wave, where neighbouring cells are transitioning from a refractory to rest state.

Rotating spiral patterns generate waves that are more frequent and regular than circular patterns (*Figure 1* 5-6 hr onwards) and hence become the dominant pattern over time, overriding the less frequent circular waves, as reported in previous studies (*Lee et al., 1996*). The mean wave frequency steadily increased towards the maximum firing frequency of cells, around once every 3–4 min (*Figure 1E*). Consequently, the proportion of spontaneously activating cells tends to zero as there is no time to activate between waves (*Figure 1D*), and the population distribution of inter-pulse intervals tends to a normal distribution (*Figure 1E*). To understand the basis of the heterogeneity in inter-pulse intervals, we quantified the single-cell differences in activation times relative to the wave centre

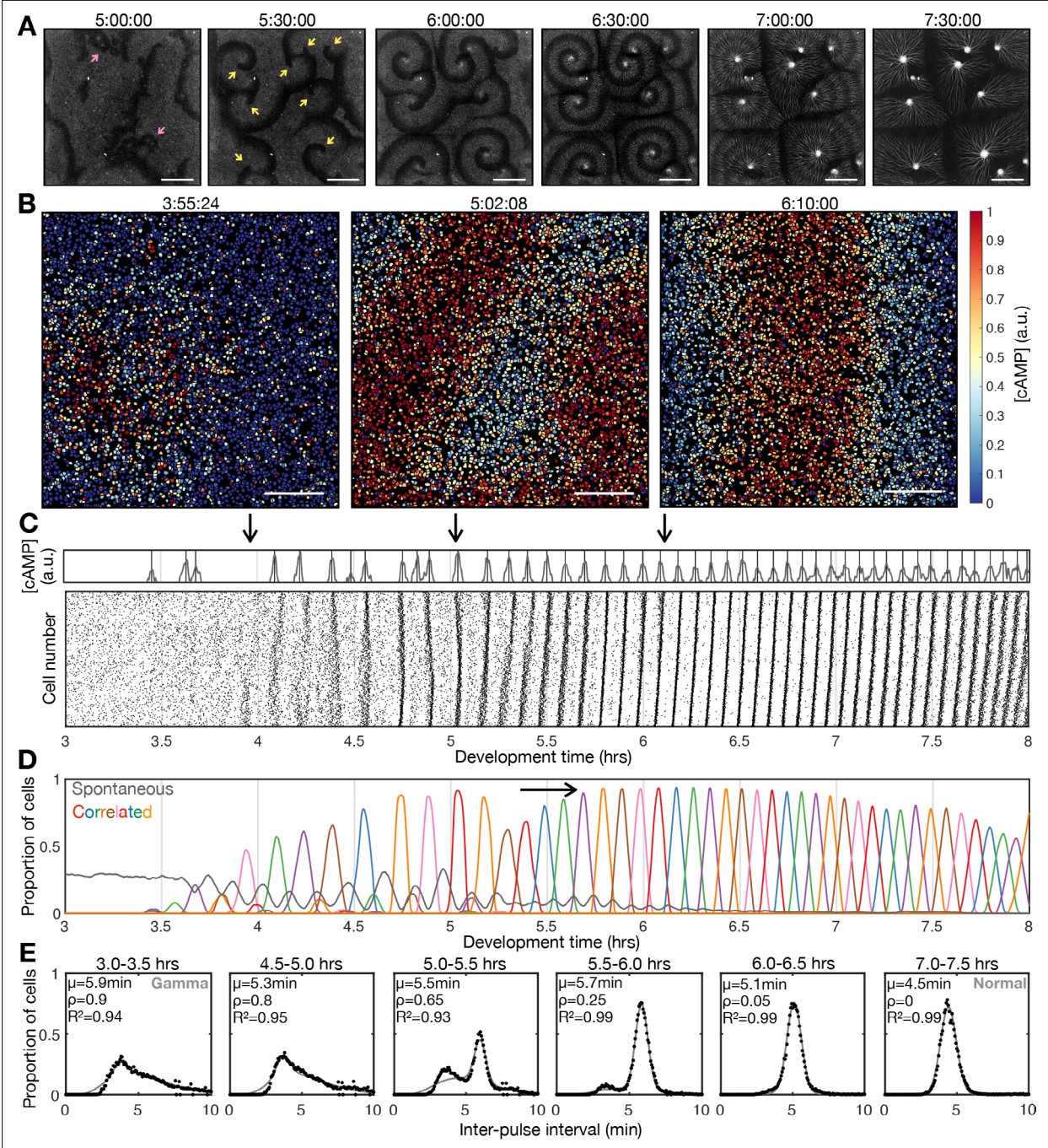

**Figure 1.** Transition to collective cell signalling. (**A**) cAMP waves (dark bands) imaged using the Flamindo2 reporter following starvation (for a representative movie see *Figure 1—video 1*). Arrows indicate circular waves (pink) and spiral waves (yellow). Scale bar: 2 mm. (**B**) The distribution of cells and internal [cAMP] (colour) following starvation. Scale bar: 200 μm. (**C**) Representative time series of the internal [cAMP] within an individual cell (top) and a plot of signal peaks for 1000 cells (bottom). (**D**) Changing proportions of correlated (coloured) and spontaneous (grey) activation events following starvation; each colour represents a single cAMP wave. Arrow: onset of spiral formation. (**E**) The distribution of inter-pulse intervals over different 30 min time intervals after starvation onset with the mean of each shown by parameter μ, together with a fitted sum of the gamma (weighted by value $\rho$) and normal (weighted by value 1-$\rho$) distribution. For the gamma distribution, the shape and rate parameter were 6.6 and 1.25 min respectively, and the mean of the normal distribution varied (from 6 to 4.5 min).

The online version of this article includes the following video and figure supplement(s) for figure 1:

**Figure supplement 1.** Transition to collective cell signalling.

**Figure supplement 2.** Formation of spiral waves via photoactivation (see *Figure 1—video 3* for additional clarity).

*Figure 1 continued on next page*

**Figure supplement 3.** Cell-cell variability in activation times.

**Figure 1—video 1.** *Dictyostelium* cAMP signalling patterns and aggregation.

**Figure 1—video 2.** Quantification of single-cell signalling and motion.

**Figure 1—video 3.** Initiation of spiral waves using optogenetics.

(*Figure 1—figure supplement 3A, B*). Aided by Markov modelling, we observed a short-term bias in firing tendencies: for example, a cell that fires ahead of its neighbours has a 60% chance of repeating this early firing, 35% chance of switching to late firing, and a 5% chance of skipping the subsequent wave (*Figure 1—figure supplement 3C, D*). Whether a cell fires before or after the wave does not affect its probability of skipping a wave. Overall, these data suggest no intrinsic difference between cells beyond a slight history dependence.

## The onset of collective migration

To relate how information encoded by spiral waves coordinates collective motion, we compared changes in cAMP wave dynamics to cell chemotaxis (*Figure 2* and *Figure 1—video 2*). Chemotaxis is directed by the external cAMP concentration field, which is determined by the release of internal cAMP (reported by Flamindo2) from individual cells. The spatial organisation and motility of cells changed rapidly, occurring around 7 hr following starvation, coincident with the dominance of spiral waves (*Figure 2A* and *Figure 2—figure supplement 1*). During this process, cAMP waves steadily became more frequent, slower and thinner, and cells became faster, more directional, and persistent in their movement (*Figure 2B and C*). The profile of the mean cAMP pulse in single cells remained constant with each wave. Since the duration of a cAMP pulse ($\tau$) remains constant, and the wave speed ($v$) decreases, then the wave width ($w = v \times \tau$) also decreases. A thinner wave generates a steeper external cAMP gradient, which may enhance chemotaxis (*Song et al., 2006*).

To quantify collective cell motion, we used the following two measures: (i) the mean chemotaxis index, the proportion of cell movement in the direction of the oncoming wave (*Skoge et al., 2014*; *Figure 2B and D*), and (ii) the local correlation of cell velocity (*De Palo et al., 2017*; *Figure 2E*). Cells consistently moved towards the wave source at wave fronts but remained spatially uniform before 6 hr of starvation due to random cell motion at the wave rear (*Figure 2B* bottom left panel). This is highlighted by a peak in the local correlation of cell motion (*Figure 2E*) and the mean chemotaxis index (*Figure 2D* bottom panel) at the wave front, both of which then decay towards zero (indicating random cell motion). For infrequent, fast and wide waves (i.e. circular patterns), the chemotaxis index and local correlation of cell motion reached their baseline values at the following wave front, indicating cells move randomly. This random cell motion disrupts the order in collective cell motion initially induced by the front of the wave. For frequent, slow and thin waves (i.e. rotating spiral waves), the chemotaxis index and velocity correlation significantly change as the cAMP wave frequency increases between 6 and 8 hr following starvation. From 6.5 hr onwards, when the wave period is less than 4.5 min, the measures of collective cell motion do not relax back to their previous values by the next wave and hence gradually increase over time to constant, elevated values across the entire wave (*Figure 2D and E*). In parallel, the cell speed at the front of the wave increases incrementally with each wave until it peaks at around 7 μm min$^{-1}$ (*Figure 2D* second panel). These data are consistent with models where decreasing the wave period causes cells to gradually become more persistent towards the direction of the wave source and hence gain consistent collective motion (*Skoge et al., 2014*).

To test the relationship between wave frequency and collective migration in the full population context, we photoactivated bPAC in a small patch of cells at different intervals and monitored the resultant activation waves (*Figure 3* and *Figure 3—video 1*). Consistent with our observations shown in *Figure 2*, cells collectively migrated towards the signalling centre only when the induced wave period was less than 4.5 min (*Figure 3A*). However, photoactivation of cells with a period less than 3.5 min did not result in aggregation. We infer that this is because cells cannot relay a steady periodic signal close to their maximum firing rate initially, resulting in a missed wave every second

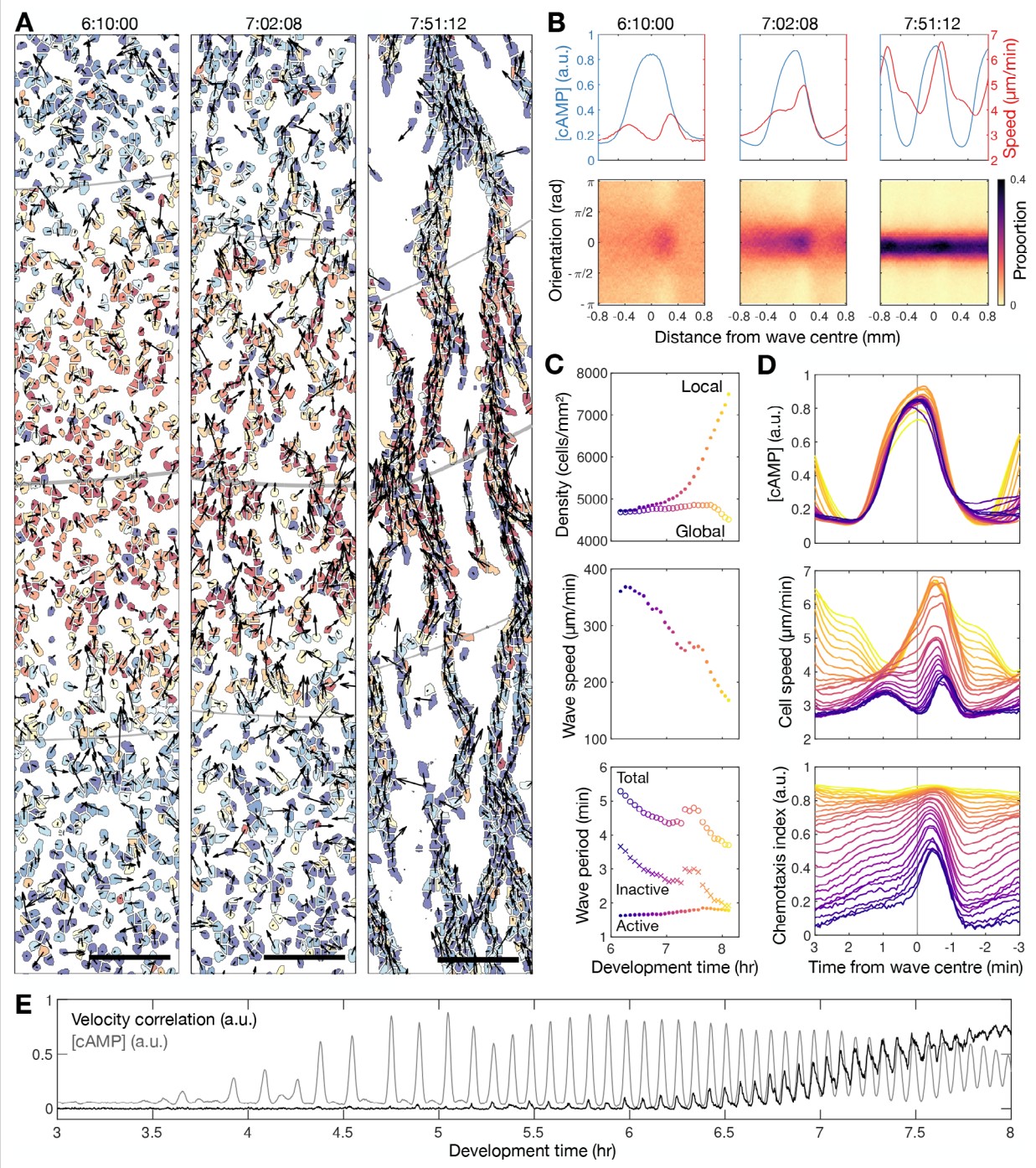

**Figure 2.** Transition to collective cell motion. (**A**) Snapshots of cell velocities (arrows) and signal states (coloured by [cAMP], as in *Figure 1B*) surrounding a cAMP wave (travelling from top to bottom) at three different time points following spiral wave formation. The inferred wave boundaries and peak are highlighted as grey lines. Scale bar: 100 μm. (**B**) Top: the mean cell [cAMP] (blue) and speed (red); bottom panels show the orientations of the cells with respect to the direction of wave motion (0 is facing the wave; ± π facing away from the wave), as a function of distance from the wave centre for the three waves shown in A. (**C**) The local (mean) and global population density (top), wave speed (middle), and wave period (bottom) for each wave during 6–8 hr following starvation. Each data point is coloured by wave number. (**D**) The mean cell [cAMP] (top), speed (middle), and chemotaxis index (bottom) across each wave, represented as the time to the wave centre (positive values correspond to behind the wave). Each line is coloured by wave number as in C. (**E**) The mean cell [cAMP] (grey) and local velocity correlation (black) between cells within a 400 μm² area.

The online version of this article includes the following figure supplement(s) for figure 2:

**Figure supplement 1.** Transition to collective cell motion.

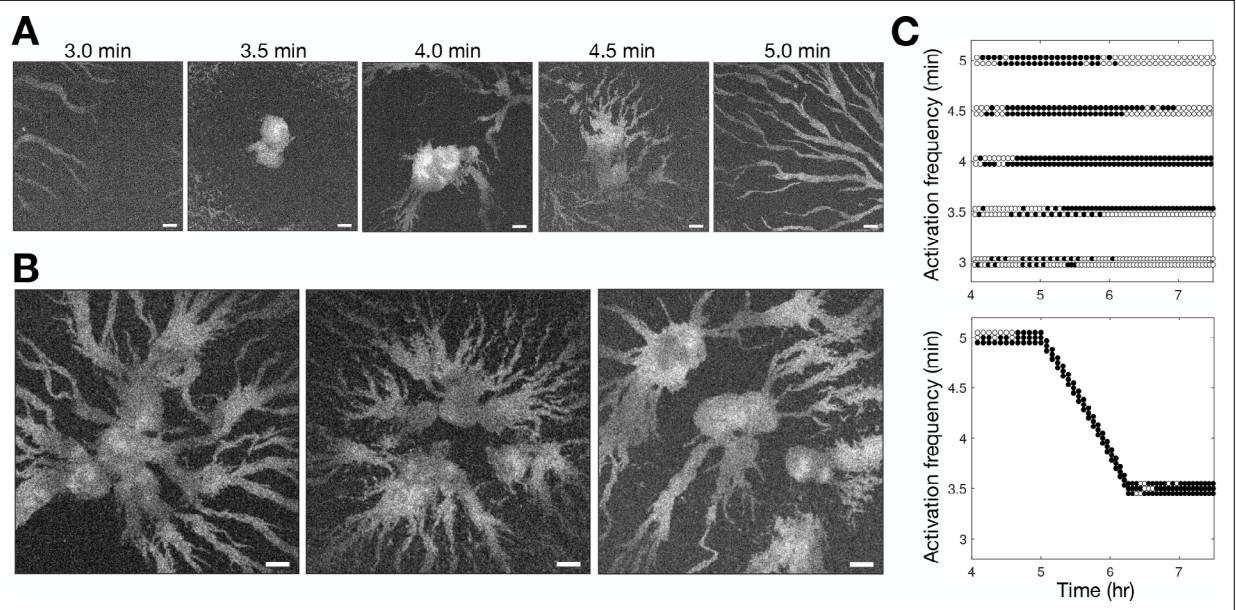

**Figure 3.** Controlling aggregation by setting the cAMP wave frequency. See *Figure 3—video 1* for live movies. (**A**) Raw images of cells subjected to 3 hr of periodic activation of the blue-photoactivable adenylyl cyclase (bPAC) (with pulses applied at centre of the field of view) at 3, 3.5, 4, 4.5, and 5 min intervals. Scale bar: 100 µm. (**B**) Same as A except that the frequency steadily increased from 5 to 3.5 min intervals. Note that the size of the images is larger than those in A. Scale bar: 100 µm. (**C**) Time series of the periodic photoactivation protocol for the experiments shown in A (top) and B (bottom). The coordinate of each circle represents the time (x-axis) and frequency (y-axis) of photoactivation, and its colour indicates whether it resulted in a relayed circular wave (black) or not (white). Experiments in A are in duplicate, and those in B are in triplicate.

The online version of this article includes the following video for figure 3:

**Figure 3—video 1.** Inducing aggregation by optogenetic control of cAMP pulsing.

https://elifesciences.org/articles/83796/figures#fig3video1

photoactivation event (*Figure 3C*), leading to the signalling centre becoming 'taken over' by another set of period waves. Motivated by the observed change in wave frequency during aggregation, we sought to test if it were possible to induce large cell aggregates by photoactivating cells at a faster rate than what we observe experimentally by reducing the wave period gradually from 5 min to 3.5 min from 5 to 6 hr following starvation, rather than 6–8 hr (*Figure 3C* second panel). This was indeed achievable, resulting in the accumulation of a significantly larger number of cells around the signalling centre and conglomerates of several aggregates (*Figure 3B*). These artificial signalling centres induce large aggregates by entraining the territories of neighbouring signalling centres. Our data support the notion that the increase in wave frequency is a crucial determinant of the transition to collective motion. We now turn to the question of how cells generate this increasing wave frequency.

## Spiral wave progression

To characterise the regulation of spiral wave dynamics, we imaged and analysed the motion and signalling of individual cells around the spiral core following the onset of spiral formation. The spiral core is the region about which the spiral tip (the end of the activation wave) moves. Based on the general theory of excitable media, the spiral tip will sustain itself and the entire spiral wave by virtue of circular motion (*Tyson and Keener, 1988*; *Winfree, 2001*). In other words, the circulation of the spiral tip is the organising source of the spiral wave. To better understand the dynamics of spiral tip circulation, and how it relates to the changes in wave dynamics, we tracked the position of the spiral tip and the shape of the spiral wave from spiral formation to aggregation (*Figure 4*, *Figure 4—figure supplement 1*, *Figure 4—videos 1–3*). To track tip motion, we mapped the [cAMP] of individual cells to a phase variable measuring the sequence of changes in the internal [cAMP] during a cAMP pulse (*Figure 4B*, *Figure 4—figure supplement 1B, C*). During a cAMP pulse, this variable increases from 0 (baseline [cAMP]) to ±π (maximum [cAMP]) and then back to 0. This approach reveals the spiral singularity — or topological defect — which is the point in space around which cells are distributed across

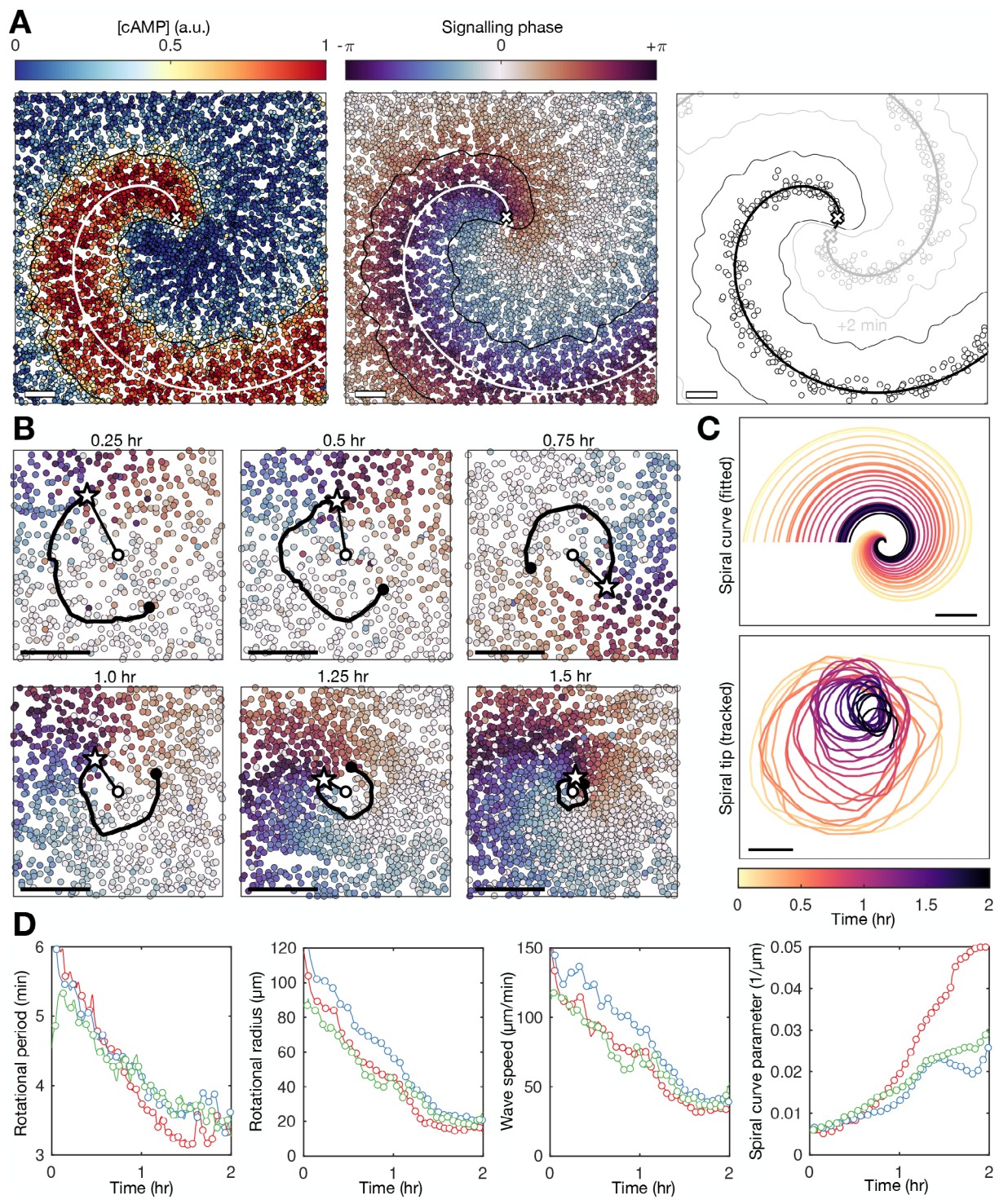

**Figure 4.** Spiral tip circulation and spiral wave progression. (**A**) The fitted spiral curve (thick line) and predicted wave boundary (thin line) with the spatial distribution of: the internal [cAMP] (first panel), the signalling phase (second panel), and maximally activated cells (phase ±π) 2 min apart (0 min – black,+2 min – grey; third panel). Scale bar: 100 μm. (**B**) 3 min tracks (thick black line) of the spiral tip (star) circulating around the pivotal axis (thick white circle) overlayed with cell positions and signalling phase. Scale bar: 50 μm. (**C**) Plots of the fitted spiral curve (mean curve per rotation; top) and track of the radius of spiral tip circulation (bottom), coloured by time since spiral wave formation. (**D**) Time series of the spiral tip circulation rotational period (first panel), rotational radius (second panel), and speed (third panel), in addition to the spiral curve parameter defining the fitted spiral shape in C (fourth panel) for each experiment (red, blue, and green) from the onset of spiral formation. Each circle represents the mean value per rotation and the lines represent the instantaneous values (e.g. rotational period is 2π /). See *Figure 4—videos 1–3* for live cell movies.

*Figure 4 continued on next page*

*Figure 4 continued*

The online version of this article includes the following video and figure supplement(s) for figure 4:

**Figure supplement 1.** Quantifying single cell signalling around the spiral core.

**Figure 4—video 1.** Quantifying signalling centre wave circulation and spiral wave structure and dynamics.

https://elifesciences.org/articles/83796/figures#fig4video1

**Figure 4—video 2.** Quantifying signalling centre wave circulation and spiral wave structure and dynamics.

https://elifesciences.org/articles/83796/figures#fig4video2

**Figure 4—video 3.** Quantifying signalling centre wave circulation and spiral wave structure and dynamics.

https://elifesciences.org/articles/83796/figures#fig4video3

the full spectrum of phase states. At each time point, we estimated the spiral tip location as the mean position of maximally activated cells closest to the singularity (*Figure 4B*). Additionally, we fitted a spiral curve to the spatial distribution of maximally activated cells (*Figure 4A and C* first panel).

Spiral waves significantly change in rotation rate and curvature during aggregation, which we refer to as spiral wave progression (*Figure 4C* and *Figure 4—videos 1–3*). These changes in spiral wave structure and dynamics are coupled to changes in spiral tip circulation (*Figure 4B–D*). Each rotation of the spiral tip generates a wave: the rate at which the activation waves circulate around the signalling centre sets the frequency of global activation waves disseminated from the signalling centre. The spiral tip rotational period decreased from 5.5 min to 3.5 min. These changes in the rotational period of the spiral tip translate to the changes in the rotation rate of the spiral wave: the decrease in the spiral tip rotation period (*Figure 4D* first panel) is consistent with the decrease in the wave period away from the spiral core that coincides with the onset of collective migration (*Figures 2C and 3*). The acceleration of spiral tip circulation is associated with spiral core contraction; the radius of circulation steadily reduced from around 100 µm to 20 µm (*Figure 4D* second panel). Meanwhile, during this contraction, the spiral tip speed decreased from 100 to 120 to 40–50 µm min⁻¹ (*Figure 4D* third panel).

How do cells at the signalling centre drive spiral wave progression? To measure the fundamental signalling parameters and how they change over time, we utilised the general theory of excitable media (*Tyson and Keener, 1988*). *Dictyostelium* cAMP waves can be shown to satisfy this theory (*Tyson et al., 1989*; *Foerster et al., 1990*). Our high-resolution datasets now make it possible to apply this theory to understand how spiral waves change during aggregation, independently of the specific equations that model cAMP release, which are still disputed (*Sgro et al., 2015*; *Kamino et al., 2017*). According to this theory, the shape of the spiral wave must satisfy a curve that is approximately Archimedean (Materials and methods, *Equation 2*). The spiral curve depends on a single parameter $k = N_0/\varepsilon D$ (referred to here as the curve parameter) which relates the signalling parameters $N_0$ (the planar wave speed), $D$ (the signal diffusion constant), and $\varepsilon$ (the ratio of the rates of cell inactivation and activation, which we call the cell activation parameter; Materials and methods, *Equation 3*). Crucially, the inverse of $\varepsilon$ is the measure of cell excitability and specifies the rate at which cells activate. By fitting the spiral curve to the spatial distribution of maximally activated cells at each time point (*Figure 4A*; *Figure 5A*, *Figure 4—videos 1–3*), throughout spiral wave progression, we obtain a measurement of how the cell excitability and planar wave speed change over time (*Figure 5B* second column). Specifically, both measures of the cell signalling phenotype can be expressed in terms of spiral wave properties that we can measure (Materials and methods, *Equation 4*), such as the normal wave speed and wave curvature (*Figure 5B* first column).

This analysis reveals that the activation parameter and planar wave speed both decrease during spiral wave progression (*Figure 5B*). Firstly, we find that the planar wave speed halves in value as the wave period decreases towards its minimum (*Figure 5D*). The theory of excitable media provides the following interpretation: as the wave period decreases, cells have less time to reset between waves, causing slower cell activation and wave speed (the dispersion relation), which gives rise to the deceleration of wave speed during core contraction. This deceleration quantitatively matches the changing wave dynamics away from the spiral core (*Figures 2C and 5B*), meaning the circulation of the spiral tip determines the wave speed away from the core, in addition to the frequency. Secondly, the theory interprets the increase in spiral wave rotation rate and curvature to be a consequence of cells becoming more excitable over time (*Figure 5C*). In other words, the spiral waves progress because

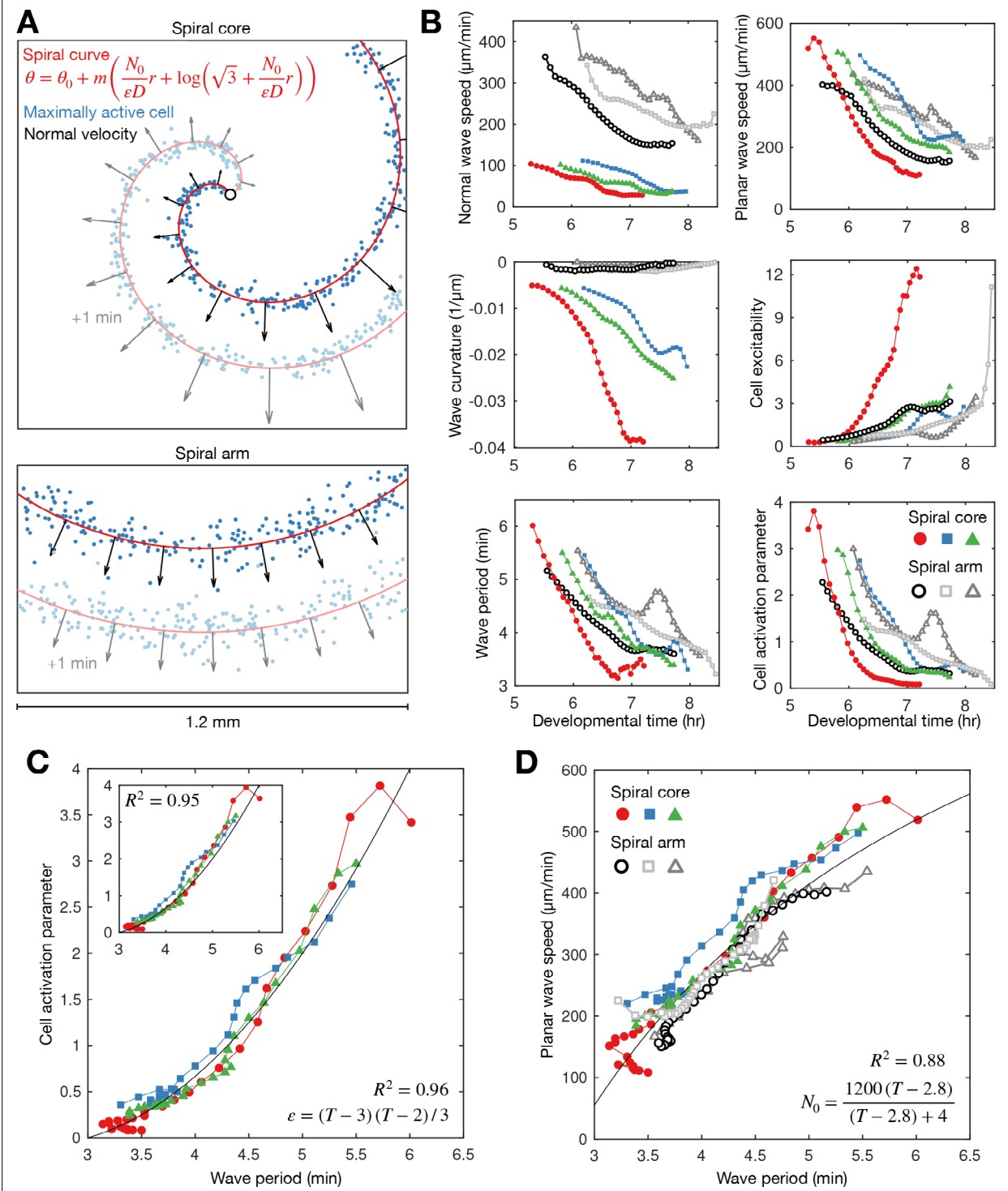

**Figure 5.** General theory of excitable media – cell excitability is coupled to wave dynamics. (**A**) Representative curve fits (red line) to the distribution of maximally activated cells (blue dots) at (top) and away from (bottom) the spiral core. The normal wave speed is shown as black arrows. (**B**) Time series of wave characteristics derived from A. Coloured lines show data from the spiral core. Greyscale lines show data away from the core. (**C**) The cell activation parameter (calculated using the eikonal relation) plotted against the wave period at the spiral core. Inset shows the same plot, but with the activation parameter derived from the curvature relation, together with the fitted quadratic function (black line). (**D**) The dispersion relation at (colours) and away from (greyscale) the spiral core, together with the fitted saturating function. The data shown in dark grey are derived from the same experiment shown in *Figure 2*. Spiral wave data is from the same experiments shown in *Figure 4* (same colours).

The online version of this article includes the following figure supplement(s) for figure 5:

**Figure supplement 1.** Mechanism of spiral core contraction.

cells can activate faster. Indeed, although there was temporal variation between experimental replicates, the activation parameter collapses onto a highly consistent curve when plotted against the wave period (*Figure 5C*), showing how the wave dynamics are strongly coupled to the excitability of the cells. A simple quadratic provides a good fit regardless of derivation used for quantifying the activation parameter (Materials and methods, *Equations 4 and 5*). In summary, this analysis implies the change in cell excitability is a key element driving spiral wave progression. We now evaluate different hypotheses for mechanisms driving spiral wave progression.

## Hypotheses

### Hypothesis 1: increased collective excitability by cell reorganisation with constant individual excitability

The existing paradigm, based on mathematical modelling, is that spiral wave progression is driven by cell aggregation (*Höfer et al., 1995*; *van Oss et al., 1996*). In this view, an increase in cell density elevates cAMP production and degradation rates per unit volume, increasing the collective excitability around the aggregate. In this way, the spiral wave rotation rate and curvature increase as cells accumulate at the spiral core, even though the intrinsic excitability of individual cells remains constant. Consistent with this paradigm, we measured a substantial decrease in the cell activation parameter (inverse of cell excitability) and increase in cell density during spiral wave progression (*Figure 6*). The relationship between density and the activation parameter is incredibly sensitive, meaning slight fluctuations in density (such as occurring through random motility) would correspond to dramatic changes in the activation parameter (*Figure 6C* first panel). However, unlike the strong relationship between wave period and cell excitability (*Figures 5C and 6C* third panel), a causative relationship between density and excitability is not supported by reliable fits (*Figure 6C* second panel). In addition, our data show that spiral wave progression occurs prior to clear changes in cell density (*Figure 6A*). Specifically, spiral core contraction and the increase in rotation rate are immediate and steady for around 1.5 hr following spiral wave formation (*Figure 6B* second panel), while the cell density sharply increases only after around 1 hr (*Figure 6B* first panel) once the wave period is below 4.5 min. Indeed, the rotational period is close to the minimum value before substantial changes in density occur (*Figure 6B* third panel). Altogether, these observations imply that an increase in cell density is not sufficient to account for spiral wave progression, at least at the onset of collective migration.

A more recent model for cAMP release (*Sgro et al., 2015*), which is based on the FitzHugh-Nagumo model rather than the Martiel-Goldbeter model (*Martiel and Goldbeter, 1987*), shows improved quantitative behaviour of cAMP release from single cells (*Huyan et al., 2021*). It is conceivable that incorporating this more recent model into the earlier models of spiral waves (*Höfer et al., 1995*; *van Oss et al., 1996*; *Dallon and Othmer, 1997*) could provide spiral wave progression consistent with our data - an increase in the spiral wave rotation rate prior to cell aggregation. To test this, we developed a hybrid partial differential equation (PDE) agent-based model (ABM; see Materials and methods and Supplementary Information) in the same vein as *van Oss et al., 1996*, but replacing the Martiel-Goldbeter model with the modified FitzHugh-Nagumo model. To explore how cellular reorganisation influences the signalling pattern, we fixed all signalling parameters such that the intrinsic cell excitability does not change. This model reproduced a rotating spiral wave and experimentally observed spatial patterning of cells (*Figure 7*, *Figure 7—videos 1 and 2*). In the model, the rotating spiral wave creates a limit cycle in cell motion (see the circular streamline in *Figure 6A* and *Figure 6—figure supplement 1*). Cells collect along the limit cycle via chemotaxis from both inside and outside the path of tip circulation, forming a ring pattern. The emergence of this limit cycle is matched in our data (*Figures 6A and 7*, *Figure 6—videos 1 and 2*). In data and model, both the radius of spiral tip circulation and the ring pattern of cells contract over time. In our model, this contraction occurred in the absence of cell-cell physical interactions (i.e. no crowding effects) but not in the absence of cell motion (*Figure 7D and E* first panel). These tests indicate that spiral core contraction can be attributed to the spatial reorganisation of cells via chemotaxis towards the circulating wave, but not necessarily due to crowding effects from cells funnelling towards the spiral core. This effect is also observed in data (*Figure 6A*): the limit cycle is 'blunted' at the location of the spiral tip, suggesting that cell chemotaxis is deflected around the external [cAMP] field of the spiral tip, inwards from the edge of the cell ring. Put simply, the tip is deflecting cells inwards.

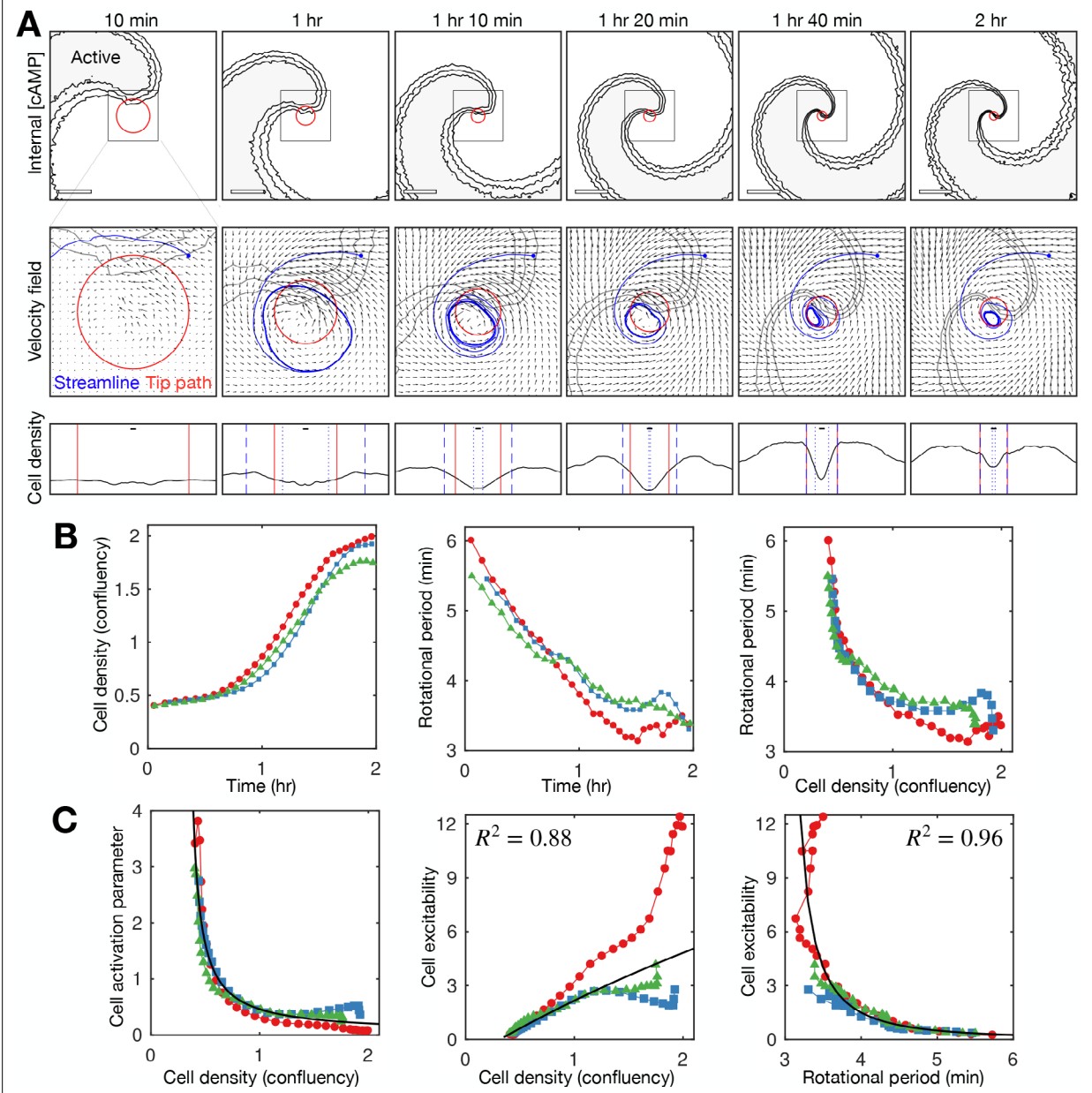

**Figure 6.** Relationship between cell density and distribution to spiral wave progression. (**A**) Top row: the mean internal [cAMP] around the tip with the mean circulation radius of the spiral tip (red). Scale bar: 200 μm. Middle row: mean cell velocity field together with a representative streamline, the mean radius of the path of spiral motion (red), and contoured outline of the spiral (grey). Bottom row: cell density together with the mean radius of the spiral tip motion (red) and the minimum (dotted) and maximum (dashed) distance of the representative streamline from the axis of spiral rotation (blue). Scale bar: 10 μm (≈1 cell diameter). Each plot represents a 5 min average. (**B**) Time series of the cell density (fraction cell confluency, estimated with cell area of $\pi\,5^2\,\mu m^2$) around the spiral core (first panel) and spiral tip rotational period (second panel), together with the relationship between the cell density and rotational period (third panel). Shown are the same three datasets (colours) shown in *Figures 4 and 5*. Data points represent average values per rotation, in contrast to *Figure 4D*. (**C**) Relationship between the cell activation parameter (first panel) and cell excitability (inverse of the cell activation; second panel) and the cell density. Relationship between cell excitability and wave period (third panel). The cell activation parameter (inverse of cell density) structured by cell density and period was fitted with a reciprocal curve ($\varepsilon = 0.04 + \frac{0.28}{n-0.32}$) and quadratic curve (same as in *Figure 5*), shown as black lines, with their respective R² values.

The online version of this article includes the following video and figure supplement(s) for figure 6:

**Figure supplement 1.** Single-cell tracks around the spiral core.

**Figure 6—video 1.** Quantifying cell velocities at the signalling centre.

https://elifesciences.org/articles/83796/figures#fig6video1

*Figure 6 continued on next page*

*Figure 6 continued*

**Figure 6—video 2.** Quantifying the mean cell velocity field.

https://elifesciences.org/articles/83796/figures#fig6video2

Since our data point to the contraction of the radius of spiral tip circulation as the driver of spiral wave progression (*Figure 4B–D*), our model suggests the following scenario for the progression of the spiral wave: as the external [cAMP] field of the spiral tip rotates about the spiral core, it draws cells from the cellular ring inwards, causing a decrease in the circulation radius hence an increase in the rotation rate. In this view, the spiral progression associated with the onset of collective chemotaxis is driven by spiral core contraction. This contrasts the original formulation where cell reorganisation increases the cell density, which increases the collective excitability.

This contraction scenario sufficiently explains spiral core contraction but not spiral wave progression. Specifically, the spiral wave does not increase in rotation rate during spiral core contraction (*Figure 7E* second and third panel). The speed of spiral tip motion decreases too much to offset the contraction in the radius of circulation (*Figure 7E* fourth panel). This is a flaw in the model since the increase in wave frequency is the key element of change coinciding with the onset of collective cell migration. By adding an additional component to the model, which allows the excitability of individual cells to increase over time, this flaw can be bypassed, with the model outputting a strong increase in wave frequency (*Figure 7—figure supplement 1* and *Figure 7—video 3*).

## Hypothesis 2: increased individual cell excitability driven by wave dynamics

An increase in cell density is not sufficient to explain spiral wave progression. This suggests that spiral wave progression does not arise because of the increase in collective excitability resulting from aggregation, but cells become more excitable individually. How would this increase in individual cell excitability occur? Based on the relationship in *Figure 5C*, we propose that individual cell excitability (i.e. the rate of cell activation) increases as the wave period decreases. That is, cell excitability is coupled to the wave dynamics. This hypothesis states that spiral wave progression is due to positive feedback between cell excitability and the activation wave period: in the context of the spiral wave, which generates periodic waves, the excitability of cells increases the rotation rate of the spiral wave, thereby decreasing the wave period, which raises cell excitability and so forth.

Due to the conservation of the rotational period at the signalling centre and wave period away from the signalling centre, one consequence of Hypothesis 2 is that cell state should be similar across the field. To test this property, we used the quadratic relationship between the cell activation parameter and wave period (*Figure 5C*) to rederive the relationship between the planar wave speed and wave period (the dispersion relation) in a different context – away from the spiral core. To measure the wave dynamics and structure away from the spiral core (*Figure 6B*), we exploited that the spiral curve approaches a circle distant from the core (*Figure 6A*) and fitted an equation for a cone to the time evolution of the spatial distribution of maximally activated cells (Materials and methods). Using this approach, we find a consistent dispersion relation both at and away from the spiral core (*Figure 5D*). In other words, the changes in cell state due to wave dynamics are the same across the population. This match was critically dependent on the relationship between cell excitability and wave period that is stated in *Figure 5C*.

To test whether the coupling of excitability to the wave dynamics sufficiently describes spiral wave progression, we modelled cAMP spiral waves at constant cell density, with the intrinsic excitability of cells changing in response to external cAMP (*Figure 8* and *Figure 8—video 1*). The earlier models analysed by *Höfer et al., 1995* and *van Oss et al., 1996* were based on the spatial continuum extension of the Martiel-Goldbeter model analysed by *Tyson et al., 1989*. The key extension to the Tyson model by Hofer/van Oss is that the cell density can change. By using the Tyson model, we return to the original formulation, but instead of allowing density to change, we couple cell excitability to the cAMP wave dynamics. We set the cell activation variable to be a simple time-dependent function where the variable tends to a baseline value and decreases in response to external [cAMP] (Materials and methods). *Figure 8A* shows the time evolution of the simulated spiral wave. This solution reproduces features of spiral wave progression such as the rate of spiral core contraction (*Figure 8B*), increase in spiral wave curvature, increase in the spiral rotation rate, and the coupling between the cell activation parameter and the wave period (*Figure 8C*).

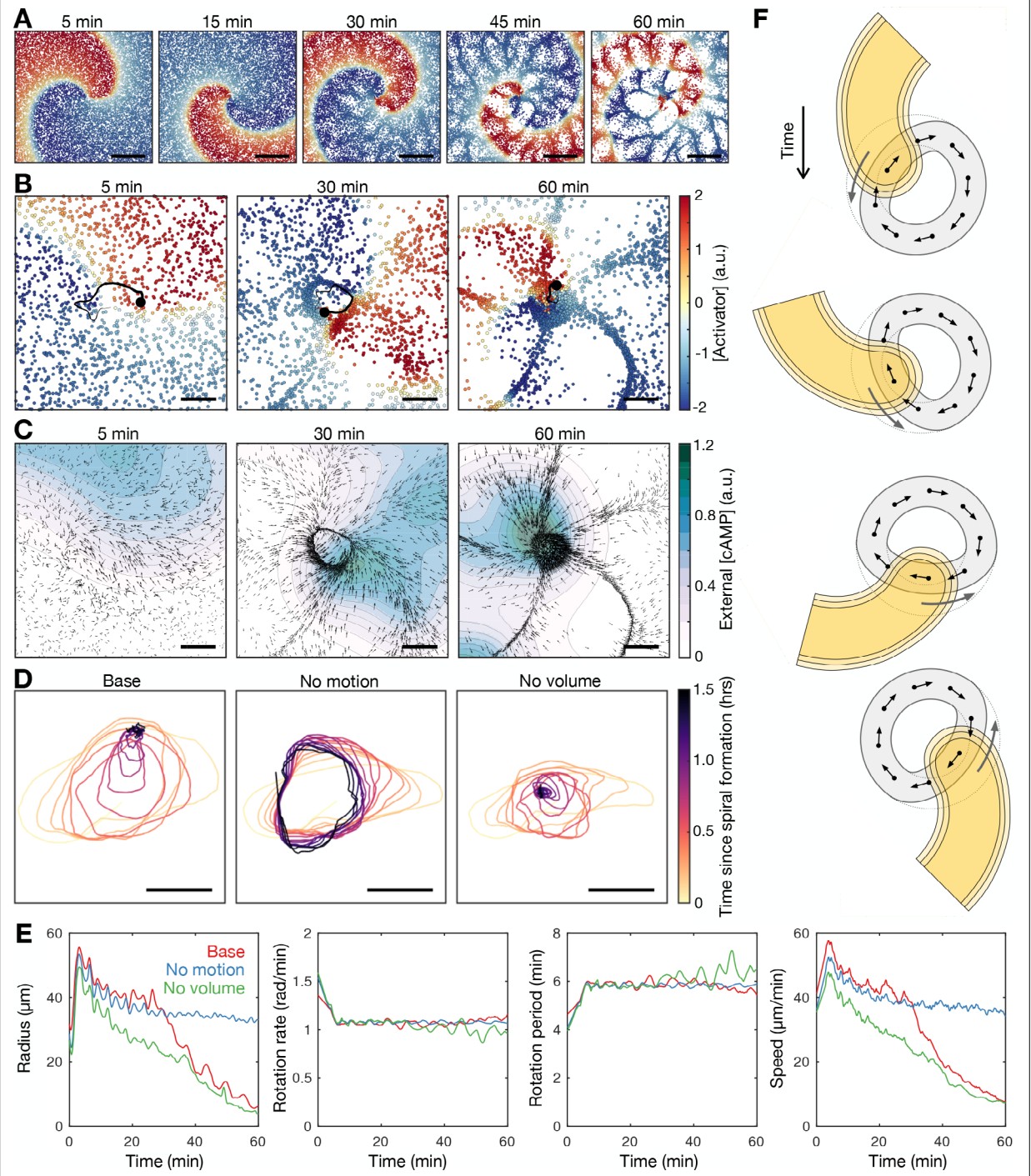

**Figure 7.** Mathematical model of spiral wave progression (constant cell excitability, variable cell density). (**A**) Simulation of the cell population distribution across space and the concentration of the species governing cAMP release (colour). Scale bar: 500 µm. (**B**) Same as A together with the track (black line) of the spiral tip (black circle). Scale bar: 100 µm. (**C**) Progression of the external [cAMP] field (colour) with cell velocities (black arrows). Scale bar: 100 µm. (**D**) Tracks of the spiral tip (coloured by time) for the original simulation (shown in A–C), a simulation without cell motion (second panel) and without cell volume (third panel). Scale bar: 50 µm. (**E**) Time series of the spiral tip circulation radius, rotation rate, rotation period, and speed for the simulated tracks presented in D. (**F**) Schematic of the mechanism of spiral tip contraction; as the spiral tip travels around the ring pattern of cells (grey), its external [cAMP] field (yellow) directs cell motion (black arrows) inwards from the ring tangent, resulting in the deformation of the cellular ring.

The online version of this article includes the following video and figure supplement(s) for figure 7:

**Figure supplement 1.** Model extension-increasing cell excitability recapitulates spiral wave progression.

*Figure 7 continued on next page*

*Figure 7 continued*

**Figure 7—video 1.** Inferring the external cAMP field.
https://elifesciences.org/articles/83796/figures#fig7video1

**Figure 7—video 2.** Modelling the external cAMP field.
https://elifesciences.org/articles/83796/figures#fig7video2

**Figure 7—video 3.** Modelling the spiral wave with constant (left) and increasing excitability (right).
https://elifesciences.org/articles/83796/figures#fig7video3

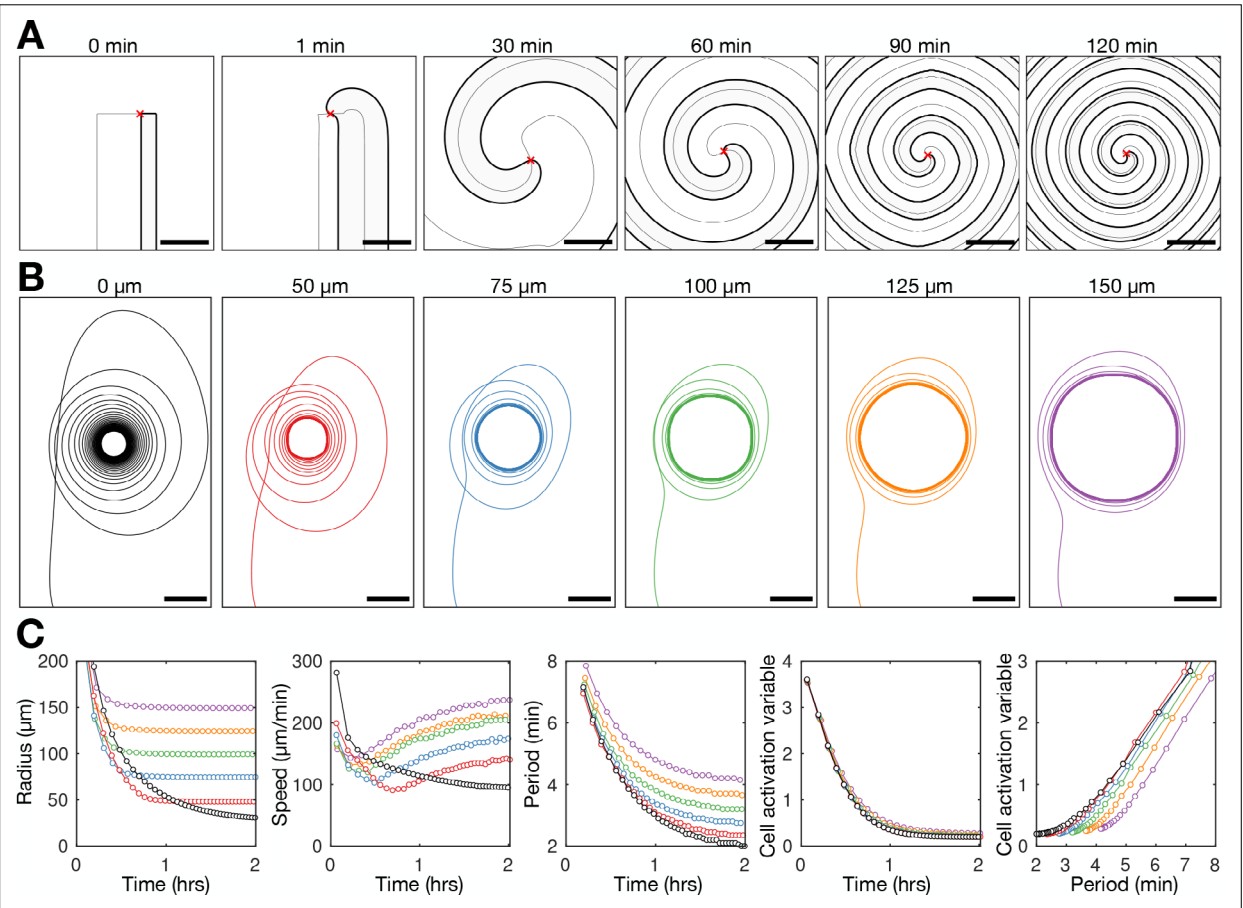

**Figure 8.** Mathematical model of spiral wave progression (constant cell density, variable cell excitability). (**A**) Numerical solution depicting the spatial distribution of the external [cAMP] (black contour with shaded interior, $u = 1$) and the inactivator species (grey contour $v = 0.6$). Also shown is the spiral tip location (red cross) estimated by the contour intersections. (**B**) Path of spiral tip circulation without (black) and with (colours) a physical block of various sizes (radii shown above). Scale bar: 100 μm. (**C**) Time series of the spiral tip circulation: radius (first panel), speed (second panel), rotation period (third panel), and the cell activation variable (fourth panel). Also shown is the relationship between the cell activation variable and wave period (fifth panel). The colours represent the measures of circulation for the differently sized physical blocks shown in B.

The online version of this article includes the following video and figure supplement(s) for figure 8:

**Figure supplement 1.** Mechanism of spiral wave progression.

**Figure 8—video 1.** Simulating spiral wave progression.
https://elifesciences.org/articles/83796/figures#fig8video1

**Figure 8—video 2.** Simulating spiral wave progression with a physical block at the signalling centre.
https://elifesciences.org/articles/83796/figures#fig8video2

The similarity between our experimental data and this model (even with parameter values rounded to the nearest order of magnitude) suggests that the positive feedback between cell excitability and the wave period is a sufficient mechanism driving spiral wave progression (*Figure 8—figure supplement 1*).

Lastly, the general theory allows us to predict the radius of spiral tip circulation from the measured cell excitability (the curvature relation, Materials and methods *Equation 5*). Is the change in cell excitability driving the change in circulation radius (Hypothesis 2) or is the radius driving the change in excitability (Hypothesis 2)? If Hypothesis 1 was true, then a change in the measured radius of circulation should precede the change in excitability and hence the predicted radius. In Hypothesis 2, the circulation radius adapts to changes in cell excitability, in line with previous experiments using caffeine, which inhibits cAMP relay (i.e. reduces excitability), caused a larger radius of circulation (*Siegert and Weijer, 1989*). Further supporting Hypothesis 2, our data show the change in predicted radius precedes the change in the measured radius, with convergence at the end of spiral wave progression (*Figure 5—figure supplement 1*). That is, the radius of spiral tip circulation is a consequence, not driver, of spiral wave progression.

Exploring the properties of the radius contraction in the adapted Tyson model shows that the inclusion of a hole in the cell density field does not hinder spiral wave progression. The spiral wave becomes pinned to the hole, which sets the minimum radius of spiral tip circulation (*Figure 8B* and *Figure 8—video 2*). The wave period reduces (*Figure 8C* third panel) because the speed of the spiral tip speed remains elevated (*Figure 8C* second panel) by virtue of an increase in excitability (*Figure 8C* fourth and fifth panel), compensating for a large radius of circulation (*Figure 8C* first panel). This simulated result contrasts Hypothesis 1, suggesting that preventing core contraction does not impede spiral wave progression.

## Hypothesis testing

We have investigated two hypotheses for spiral wave progression. The prevailing paradigm (Hypothesis 1) is that excitability increases due to an increasing cell density; however, this model is not supported by our data comparing density with wave period and excitability (*Figure 6*). Our reformulation of this hypothesis predicted that cell rearrangements drive the increase in the spiral wave rotation rate via contraction of spiral tip circulation (*Figure 7*). In contrast, Hypothesis 2 proposes that the wave dynamics change cell excitability (*Figure 8*). The key difference between these hypotheses is that in Hypothesis 1, spiral tip contraction is the driver of spiral progression (*Figure 7F*), whereas in Hypothesis 2, contraction is a consequence (*Figure 8—figure supplement 1*). Exploring the properties of the models underlying these hypotheses provides a clear experimental test: to constrain the radius of spiral tip circulation.

To experimentally test the competing hypotheses, we imaged *Dictyostelium* aggregation in the presence of physical blocks (low melt agarose particles with radii ranging between 25 and 150 μm; *Figure 9* and *Figure 9—video 1*). Spiral waves can become pinned to the blocks (*Figure 9A*), with their minimum radius of circulation set by the block radius (*Figure 9B*). Spiral waves that are pinned to a block cannot fully contract (*Figure 9C* first panel) but nevertheless undergo a steady increase in rotation rate and hence decrease in wave rotational period (*Figure 9C* third panel). The circulation rate increases because the spiral tip speed does not decrease to the same extent as the unconstrained spirals (*Figure 9C* second panel). This observation is in agreement with the model simulations shown in *Figure 8*, revealing the same effects as in the data – wave speed increased to compensate for larger radius of circulation, generating a reduced rotational period. This observation also reflects our simulations predicting that spiral waves can progress in the presence of physical blocks due to a coupling between the cell excitability and wave dynamics. In addition, this coupling between wave frequency and excitability is also apparent in our photoactivation experiments, in which a gradually increasing pulse frequency enhanced aggregate formation (*Figure 3*). Together, these observations are consistent with Hypothesis 2, but not Hypothesis 1: the contraction of the spiral tip circulation is a consequence, not a driver, of spiral wave progression.

Overall, our approach provides a clear interpretation of our data that spiral wave progression results from positive feedback between cell excitability and wave frequency, arising only in the context of the spiral wave pattern (*Figure 8—figure supplement 1*). That is, from the initial symmetry breaking event (*Figure 1—figure supplement 2*), the formation of a rotating spiral wave produces

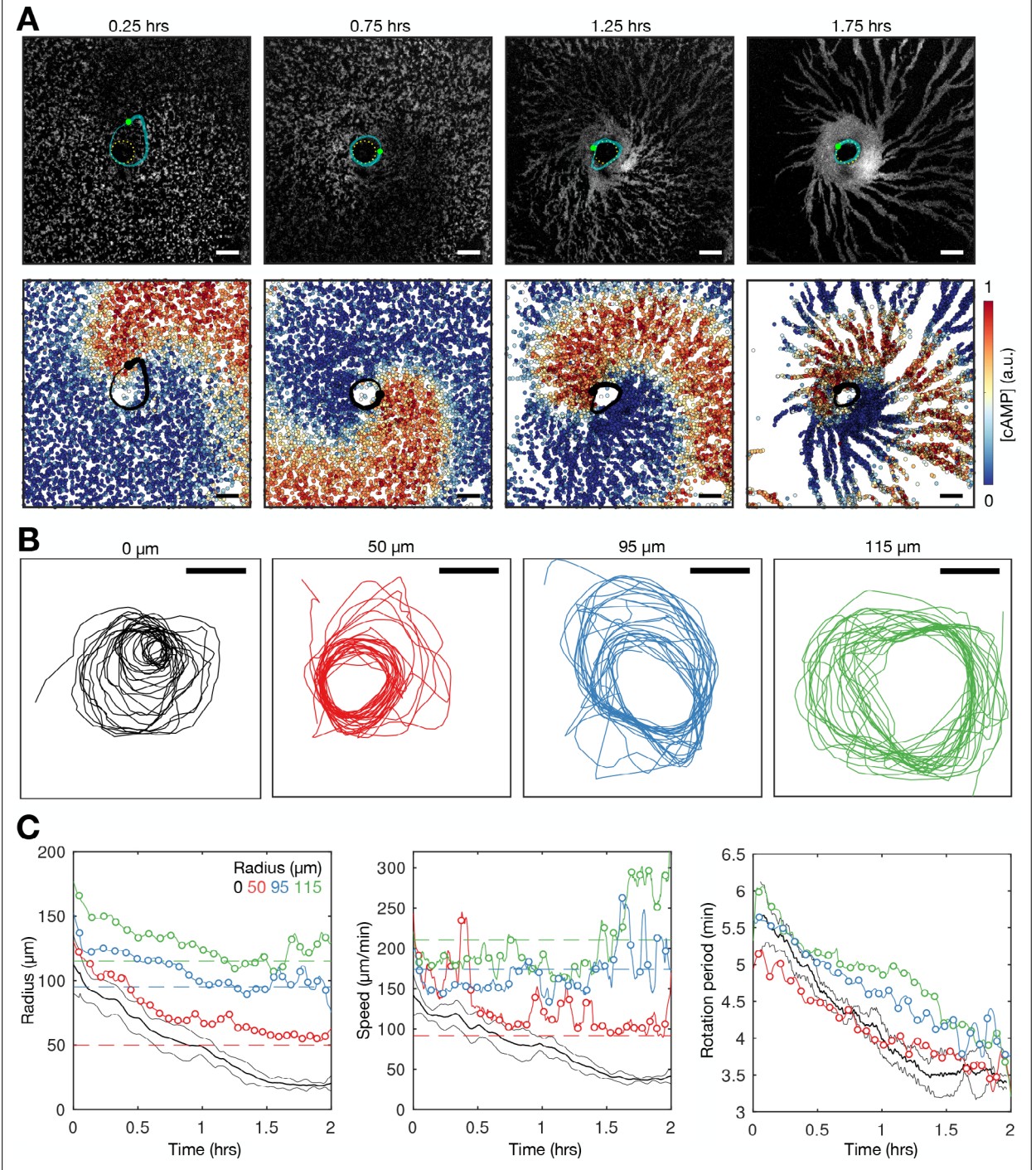

**Figure 9.** Physical occlusion of spiral core. See *Figure 9—video 1* for live cell movie. (**A**) Top: raw images of the population distributed around a physical block (dashed circle) with the spiral tip (green dot) and its track (blue line). Bottom: the spatial distribution of the internal [cAMP] with the spiral tip (black) dot and its track (black line). Scale bar: 100 µm. (**B**) Tracks of the spiral tip position for physical blocks of various sizes (radii shown in titles). The control case (no physical block, black) is from a dataset shown in *Figures 3–5*. (**C**) The spiral tip circulation radius (first panel), speed (second panel), and rotation period (third panel) around physical blocks with different average radii (dashed lines). Also shown is the mean (three experiments) of the control data without blocks shown in *Figures 3–5* (black). Panel 2 also shows the predicted speed (dashed lines) corresponding to the maximum rotation rate 3.5π/6 around the block radius.

The online version of this article includes the following video for figure 9:

**Figure 9—video 1.** Spiral wave progression with a physical block at the signalling centre.
https://elifesciences.org/articles/83796/figures#fig9video1

stable periodic activation waves that increase cell excitability, then increases the spiral rotation rate and hence wave frequency, which then feedback onto further increases in cell excitability.

## Destabilisation of the spiral wave

To conclude, we now describe the cAMP wave dynamics following aggregation. As development proceeds, more and more cells move into the aggregate. Previous studies show the activation waves are organised as a multi-armed spiral in aggregates (*Siegert and Weijer, 1995*). How does this continued cell accumulation affect the cAMP dynamics at the signalling core and, consequently, the encoding of information by cAMP waves?

To characterise the single- to multi-armed spiral transition, we monitored the structure and dynamics of the activation waves following ring collapse (*Figure 10*, *Figure 10—figure supplement 1* and *Figure 10—video 1*). Single-armed spirals destabilise only once the cellular ring fully contracts. Contraction results in a phase singularity (convergence of all signalling phases) that is fixed at the aggregate centre (*Figure 10—figure supplement 1A*). The phase singularity and spiral wave are unstable and disintegrate over time. This is likely because the cells are not infinitely small and can only be in one signalling phase at a time. In other words, the cells at the centre cannot activate at a rate necessary to support the full spectrum of signalling phases. The destabilisation process begins by cells signalling out-of-phase with the spiral wave, resulting in the emergence of transient patches of cell activation that gradually increase in number, size, and lifetime (*Figure 10*, around 2 hr following spiral formation). These activation patches rotate about the aggregate while fusing with, and breaking apart from, other activation patches and the original spiral wave, which progressively disintegrates (*Figures 10A and 2–3* hr following spiral formation). This process concludes with the appearance of three stable, slow-moving and uncurved activation waves that rotate about the same fixed point at the aggregate centre (*Figure 10A*, from 3 hr following spiral formation). At this point, the signalling regime can be classed as a multi-armed spiral. Although the ring has collapsed, the centre of the aggregate contains relatively few cells (*Figure 10—figure supplement 2*). At this point, there is no phase singularity: the few cells that occupy this central region appear to have signalling phases that are uncorrelated with each other and the surrounding multi-armed spiral wave (*Figure 10—figure supplement 1* – last panel). This central region of uncorrelated signalling phases is characteristic of theoretical chimera spiral waves (*Totz et al., 2018*).

To analyse the transition from the single- to multi-armed spiral, we generated a set of kymographs of cell activation states across the aggregate: one that is cross-sectional and the others that are circular (*Figure 10B*). The cross-sectional kymograph shows a disordered signalling regime between 2 and 3 hr following spiral formation, from around 45 min after cellular ring closure. In the circular kymographs, both the single- and multi-armed spiral waves appear as diagonal lines across space and time (<2 and >3 hr), meaning a constant rotation rate. These data show that the single-armed spiral (<2 hr) has a rotational period of 3.4±0.2 min (the minimum cell firing period, see *Figures 1, 3 and 4*), which is three times faster than the three-armed spiral whose rotational period is 10.1±0.7 min and wave period is 3.3±0.5 min. This means that the overall frequency with which a cell experiences a wave is the same because the three-armed spiral provides three waves in its longer rotation period. As such, the frequency of cAMP waves disseminated from the signalling centre, along the streams, remains the same for both signalling patterns. That is, both patterns encode the same temporal information.

Slow- and fast-waves coexist at the outside of the aggregate midway through the transition to the multi-arm spiral. This is visualised as a mixture of lines of two different gradients in the kymograph (*Figure 10B*, fourth panel, 2–3 hr). This observation suggests that both signalling regimes coexist by the mixing of cells that relay either the single- or multi-armed spiral wave. The multi-armed spiral wave eventually becomes dominant as the proportion of cells that relay the single-armed spiral wave declines. Altogether, our data suggest that the signalling pattern adapts to the change in aggregate geometry while maintaining an active phase wave that generates waves at the maximum frequency.

## Discussion

To determine the properties of signal coding during signal relay, we have carried out a characterisation of collective signalling and cell motion during *Dictyostelium* morphogenesis. Our analysis shows how cell aggregation is coordinated by the dynamics of circulating waves of cell activation at the

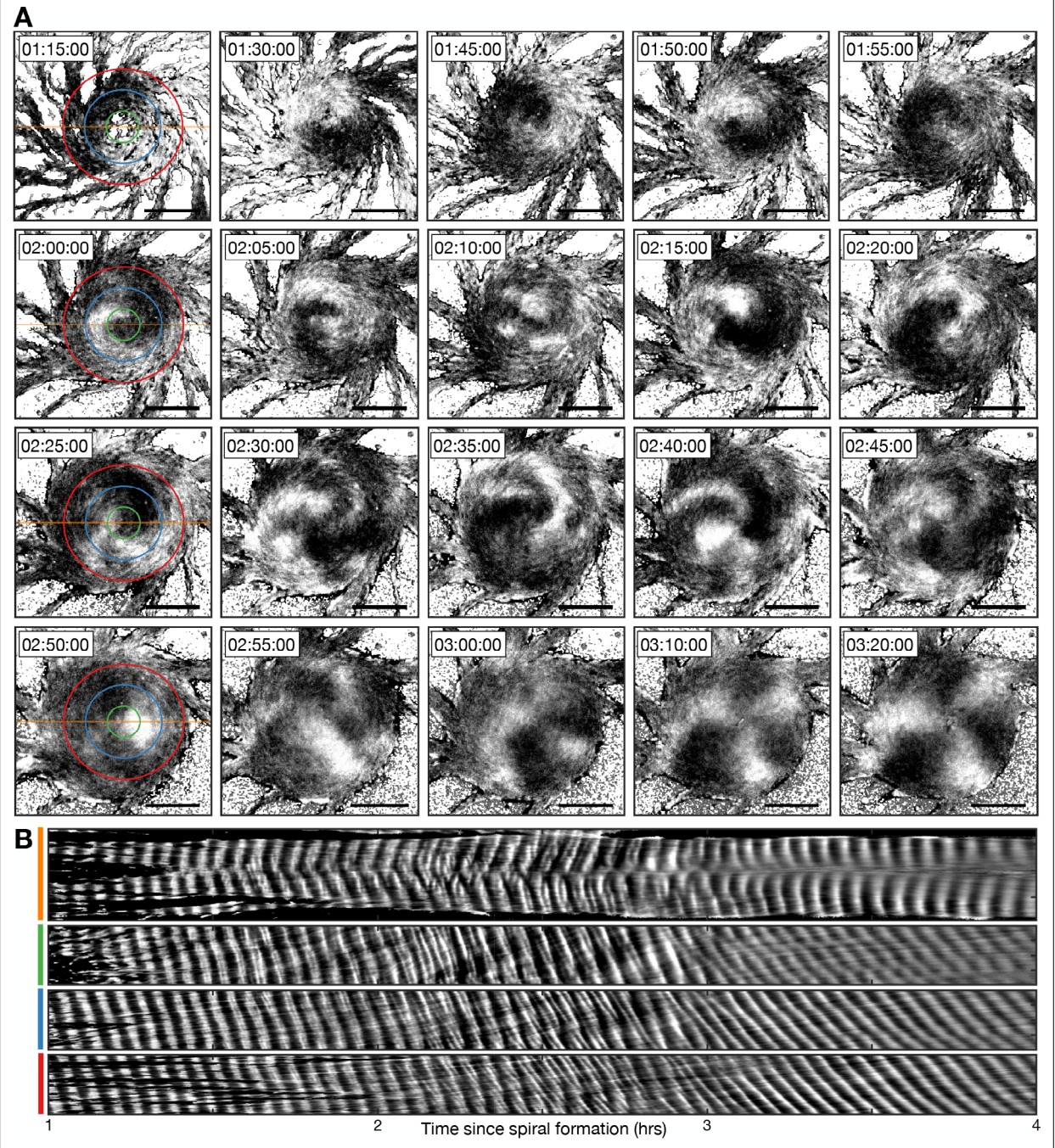

**Figure 10.** Transition from the single- to multi-armed spiral wave. (**A**) Representative intensity-normalised images of the internal [cAMP] (dark - high, light - low) distributed across cellular aggregates at various time points (labelled) following spiral formation. The images in the left column are annotated with the curves from which the kymographs in B are derived. Scale bar: 100 μm. (**B**) Kymographs of the data presented in A along a cross-section through the aggregate centre (yellow) and circles with radius 30 μm (green), 70 μm (blue), and 110 μm (red).

The online version of this article includes the following video and figure supplement(s) for figure 10:

**Figure supplement 1.** Transition from the single- to multi-armed spiral wave.

**Figure supplement 2.** Transition from the single- to multi-armed spiral wave.

**Figure 10—video 1.** Transition from the single- to multi-armed spiral wave.

https://elifesciences.org/articles/83796/figures#fig10video1

signalling centre. The circulation rate modulates the speed, frequency, and width of long-range periodic activation waves, spatially organised as a spiral. Leading up to aggregation, the circulation rate of the spiral tip gradually increases, resulting in an increase in the global wave frequency and a decrease in wave speed by virtue of the spiral wave rotating faster and becoming more curved. Using the general theory of excitable media, mathematical modelling, and physical perturbations of population structure, our data imply the increase in the circulation rate arises due to positive feedback between the cell signalling pattern and cell signalling state. Specifically, the wave circulation rate is determined by the ability of cells to relay the signal (cell excitability), whilst the ability to relay increases as the circulation rate increases. As such, the information encoding a morphogenetic transition unfolds from positive feedback between the signalling pattern and the cell signalling properties at the signalling centre. Altogether, our results describe how the collective behaviour of $10^5$–$10^6$ cells across several millimetres is organised by circulation of a self-sustaining signal over $10^2$–$10^3$ cells.

This study distinguishes biological activation waves from those produced by idealised mathematical models and chemical reactions (*Winfree, 1972*). The difference is that biological activation waves can encode information that can be modulated and translated to changes in the spatial organisation and gene expression of individuals beyond the spatial scale of the individual, which can feedback to changes in activation wave dynamics and structure. For *Dictyostelium* aggregation, information is generated locally by the rate of circulation of the spiral tip and decoded globally by cell chemotaxis. Here, by information, we mean the properties of the periodic waves, such as frequency and speed, which are translated by cells into a chemotactic and/or gene regulatory response. Information decoding via cell chemotaxis has been studied at both the single cell and population levels. There are several contrasting views regarding the wave properties which cells interpret for chemotaxis. One view is that the ability for cells to move collectively increases with an increase in wave frequency. *Skoge et al., 2014* showed, by controlling the frequency of cAMP waves, that the chemotactic index increased as the frequency became maximal (*Skoge et al., 2014*). This observation is consistent with both our correlative and optogenetic data showing onset of collective migration as the wave frequency approaches the maximum cell activation rate. An alternative view is that cells 'count' waves: artificial rotating waves revealed that 4–5 waves at high frequency can induce cell polarity migration (*Nakajima et al., 2016*). While we cannot rule out a counting mechanism, based on our data here and early data on population measurements of aggregation, in which cells would be counting upwards of 30 waves before making the decision to collective migrate, any counting mechanism would be context-dependent, for example, using counts above a certain frequency of pulses. An alternative view is based on the wave gradient, where cells move maximally towards steep stationary (*Fisher et al., 1989*; *Song et al., 2006*) and/or increasing (*Varnum et al., 1985*; *Wessels et al., 1992*; *Nakajima et al., 2014*) gradients over time. Again, this view is consistent with our data, where both the wave speed and width decrease as the wave frequency increases, meaning that cells are exposed to a steeper gradient for a longer time. Overall, these different mechanisms are looking at different properties of the wave. What our study shows is that all the wave properties: speed, frequency, and width are specified by the circulation of the spiral tip. Specifically, once the frequency is set by the tip, all other features are set by default. Altogether, our study implies that the circulation rate of the signal at the signalling centre is the mechanism of information production, regardless of the specific information-decoding mechanism.

The mechanism by which cell state changes in response to the changing wave dynamics is explained, at least in part, because information contained within periodic waves is also decoded in terms of gene expression. In *Dictyostelium,* and many other systems, gene expression is responsive to the frequency of signalling. Expression of *pdsA*, which encodes a cAMP phosphodiaterase, and *csaA*, which encodes a cell-cell adhesion protein, is sensitive to the changing frequency of cAMP pulses (*Masaki et al., 2013*; *Corrigan and Chubb, 2014*). The *csaA* gene is induced during infrequent cAMP signalling, then inactivated as the pulse frequency increases as cells aggregate. The transcription factor GataC displays rapid shuttling between the nucleus and cytoplasm in response to external cAMP oscillations (*Cai et al., 2014*). This process is low-pass filtered, with nucleocytoplasmic shuttling occurring only when external cAMP oscillations reach a threshold frequency. Here, we describe a mechanism controlling a gradual increase in the global wave frequency. In the light of these findings, it is conceivable that *Dictyostelium* use high- and/or low-pass filtered transcription factors to exploit the gradual increasing wave frequency to control the timing of gene activation. In this way, the sequential

timing of activation of several transcription factors across the cell population can be coordinated by signalling centres that disseminate periodic waves that steadily change frequency over time. A change in cell excitability could also arise independent of changes in gene expression. For example, we can imagine a scenario in which the degradation of cAMP cannot keep up with the increased wave frequency. In this situation, the baseline cAMP increases over time.

In *Dictyostelium*, the temporal coding of cAMP waves allows cells to sense the extent of cell aggregation through the frequency of activation waves produced by the aggregate, which increases as aggregation progresses. In other words, the temporal code allows for long-range sensing of aggregate progression, which can be interpreted as a non-local quorum sensing mechanism. A potential advantage of this adaptive information coding system is that by moving only towards highly frequent and consistent cAMP waves, effort is then concentrated on migration directed only towards an established signalling centre. If cells were to move maximally towards each initial wave, they would meander and unnecessarily expend energy while cAMP waves settle to a stable pattern, in addition to missing out on the opportunity to chemotaxis towards sources of nutrition.

Our data reveals the transition from the single- to multi-armed spiral wave during aggregate progression (*Agladze and Krinsky, 1982*). In the established aggregate, the single-armed spiral is associated with an unstable signalling phase singularity at the aggregate centre. The multi-armed spiral lacks a clear singularity and instead is associated with relatively small numbers of desynchronised cells at the spiral core. This is the hallmark of the chimera spiral state, a phenomenon yet to be observed in a biological system (*Totz et al., 2018*). Overall, we view the transition to the multi-armed spiral as an adaption to the change in aggregate geometry to maintain an active phase wave at the signalling centre that generates activation waves at the maximum frequency.

In their physiological context, most cell populations, including *Dictyostelium*, are structured in three spatial dimensions. It is not trivial to extrapolate our findings (in the standard 2D plane context) to understand three-dimensional activation waves, owing to the limitless number of self-sustaining wave patterns—the scroll waves—afforded by the extra spatial dimension (*Siegert and Weijer, 1992*; *Winfree, 2001*; *Durston, 2013*). To better understand how biological signals are assembled, sustained, and controlled in a more general class of tissues, it would be important to extend the experimental, computational, and mathematical approaches outlined in this study to characterise the structure, dynamics, and biological effects of cAMP wave patterns that coordinate *Dictyostelium* morphogenesis in 3D.

## Materials and methods
### Generation of cell lines

*Dictyostelium* AX3 cells were engineered to express Flamindo2 and H2B-mCherry from safe-haven genomic locations for stable and uniform expression. Flamindo2 is a cAMP biosensor (*Odaka et al., 2014*; *Hashimura et al., 2019*) with a fluorescence intensity that is inversely proportional to internal cAMP levels. A codon-optimised sequence encoding the Flamindo2 reporter was knocked-in to the a*ct5* locus for stable and uniform expression (*Paschke et al., 2018*; *Tunnacliffe et al., 2018*). H2B-mCherry, knocked into the *rps30* locus, labels the nucleus for cell tracking at highcell densities (*Corrigan and Chubb, 2014*). For optogenetic activation of cAMP signalling, we used the light-activatable adenylyl cyclase, bPAC (*Stierl et al., 2011*). A codon-optimised bPAC cDNA was cloned into the extrachromosomal expression vector, pDM1203, then transfected into the *act5*-Flamindo2 cells.

### Cell handling

Cells were grown at 22°C in 12 ml of Formedium HL5 medium supplemented with penicillin and streptomycin in 10 cm tissue culture dishes. Cultures were diluted to 25% of their density each day to stop cell entry into stationary phase. Cells were not used beyond 9 days of culture. For development, log-phase cells at 80% confluency were washed and resuspended in KK2 (20 mM $KPO_4$ pH 6.0) and then plated on 1.5% agar at between $4 \times 10^3$ and $5 \times 10^3$ cells $mm^{-2}$. After 1 hr in a humidified chamber, a 1.2 cm × 1.2 cm slab of agar was cut and gently placed cell-face-down onto a glass imaging dish (Bioptechs deltaT) and then immediately submerged in silicon oil to prevent desiccation. To incorporate

physical blocks into the cell aggregation field, we sprinkled low-melting point agarose powder onto the KK2 agar bed before the agar had set.

## Cell imaging

Cells were imaged at 22°C on a custom-built inverted wide field microscope (Cairn Research) with 470 nm and 572 nm LED light sources (*Miermont et al., 2019*). This was equipped with a Prime 95B CMOS camera (Photometrics) and 10× UplanFL N objective (Olympus). A 1.2 mm × 1.2 mm area was imaged at resolution 1 μm² per pixel every 4 s for 8 hr from 2 hr after starvation. Frame stitching was used to expand the effective field-of-view size. To image cell motion and signalling about the spiral core, including the experiments with the physical block, we imaged a 1.2 mm × 1.2 mm area for 2 hr following from 5 to 6 hr after starvation. For this, we monitored cAMP wave patterns in real time and identified waves that had started to rotate about an axis.

## Photoactivation

For light activation of cAMP signalling, we used bPAC-expressing *act5*-Flamindo2 cells and a 3i spinning-disk confocal microscope with a 10× objective and Prime 95B CMOS camera (Photometrics). To initiate periodic cAMP waves, cells contained within a circular area with a ≈100 μm diameter were pulsed with 488 nm light for 1 s, at intervals to mimic physiological cAMP pulsing. Flamindo2 images (excitation 515 nm) were collected throughout the pulse sequence, every 5 s. To initiate a rotating spiral wave, we repeated the photoactivation protocol above, except that cells were photoactivated at operator-controlled points in time and space. We monitored waves in real time and identified periodic waves away from the signalling centre (around one per 5 min, 5 hr following starvation). To induce circular wave patterns, we photoactivated cells around 3 min after the previous wave. To induce asymmetric wave patterns (resulting in the broken and hence spiral wave), we photoactivated around 2 min after the previous wave.

## Cell tracking and measuring cell signalling states

The following image analysis pipeline was implemented in Matlab 2020b. To track the position of individual cells from image sequences, the IDL particle tracking algorithm (*Crocker and Grier, 1996*) was applied to the cell nuclei channel (H2B-mCherry) of the image time series. The velocity of each cell at each time point was estimated by the first derivative, second order, central finite difference of cell position. To measure the mean fluorescence intensity of each tracked cell, watershed segmentation (the Fernand Meyer algorithm) seeded from the tracked cell positions was applied to the cell body channel (Flamindo2). To infer internal cAMP levels (non-dimensional), the tracked mean fluorescent intensity of each cell was normalised and smoothed using Matlab's Savitzky-Golay finite impulse response smoothing filter. Two measures were used to identify cAMP pulses: (i) the similarity between the normalised intensity and a square pulse of length 2 min and (ii) the signalling phase $\varphi$. Variable $\varphi$ was estimated by stratifying the time series of the internal cAMP level $c$ by its time derivate $\dot{c}$ (estimated by the first derivative, second order, central finite difference). The value of $\varphi$ is set as the angle around the point $c = c^*$ and $\dot{c} = 0$:

$$\varphi = 2 \, \tan^{-1} \frac{c - c^*}{c + \sqrt{(c - c^*)^2 + c^2}}.$$

In addition to the time series of the inferred cAMP levels, we recorded the time points of the maximal cAMP level during each pulse, and the time points of cell activation ($\dot{c} > 0$) and deactivation ($\dot{c} < 0$), measured by the time point where the cAMP levels are half-maximal.

To quantify the internal [cAMP] signalling and phase field following aggregation, we tracked the fluorescence intensity of each individual pixel as a function of time, over the entire 2D image sequence. The fluorescence intensity time series was oscillatory and typically increased over time due to cell crowding. We used Matlab's Savitzky-Golay finite impulse response smoothing filter to separate the oscillatory and background component of the signal. The background component was used to estimate the cell density of the aggregate. The oscillatory component was used to estimate the spatial distribution of signalling states and phases across the aggregate, using the same technique described above.

## Activation wave curve fitting

To quantify the evolving dynamics of the spiral wave from cell track data, we first estimated the location of the spiral phase singularity at each time point as the central point of a circular region of radius 100 μm with the largest variance in internal [cAMP] across each cell. The location of the spiral tip was estimated by the mean position of five maximally activated cells closest to the location of the phase singularity. We estimate the axis of rotation at each time point by fitting a circle to the track of the spiral tip, half a revolution backwards and forwards in time. Using the Matlab function 'fit', we fit the spiral curve given by *Equation 2* (see below) to the spatial distribution of maximally activated cells, starting from the axis of spiral tip circulation (*Tyson and Keener, 1988*). To achieve this, we determined the magnitude ($r$ in *Equation 2*) and orientation ($\theta$ in *Equation 2*) of the vector extending from the axis of spiral rotation to the location of each maximally active cell. This fit provides the spiral orientation ($\theta_0$ in *Equation 2*) and curve parameter ($k$ in *Equation 2*).

Away from the spiral core, the time and place of activation for each cell were clustered in three dimensions (two spatial and one temporal) using the DBSCAN (density-based clustering of applications with noise) algorithm as a means to identify cells with correlated signalling states (e.g. during the propagation of cAMP waves) and also, by exclusion, the cells that activate independently. This approach reveals the clusters of cell activation associated with each cAMP wave. For each wave, we estimated the dynamic curve of the wave by fitting an equation for a cone ($t - t_0 = N r$, $r = \sqrt{\left(x - x_0\right)^2 + \left(y - y_0\right)^2}$, where $t_0$ and $\left(x_0, y_0\right)$ is spatiotemporal origin of the wave) to the time evolution of the spatial distribution of maximally activated cells. The slope of each of the cones fitted to the spatial distribution of maximally activated cells at each time point was used to deduce the speed of each wave.

## Quantifying cell motion

To estimate the mean cell velocity field and density around the spiral wave, we measured the mean velocity and total number of cells at regular lattice areas (step size 10 μm) in the reference frame that rotates with the fitted spiral parameter $\theta_0$ (*Figure 6—video 2*). The streamlines across this field were analysed using the Matlab function 'streamlines'.

To analyse the behaviour of single cells during stable periodic waves, the fitted cone equations were used to estimate the mean internal [cAMP] and velocity of each cell as a function of distance from the wave centre. This allows an estimation of measures such as the angle difference between the direction of wave travel and cell motion, for each cell:

$$\cos^{-1}\left(\hat{\boldsymbol{v}}_i \cdot \hat{\boldsymbol{w}}\left(\boldsymbol{X}_i\right)\right)$$

where $\boldsymbol{v}_i$ is the normal velocity of cell $i = 1, \ldots, N$, and $\boldsymbol{w}\left(\boldsymbol{X}_i\right)$ is the normal velocity of the cAMP wave at the location of cell $i$, $\boldsymbol{X}_i$. The following two formulae were used to measure collective cell motion: (i) the mean chemotaxis index (the proportion of cell movement in the direction of the oncoming wave) as a function of distance from the wave:

$$\frac{1}{N_{\Omega_j}} \sum_{i \in \Omega_j} \hat{\boldsymbol{v}}_i \cdot \hat{\boldsymbol{w}}\left(\boldsymbol{X}_i\right)$$

where $\Omega_j$ represents the subset of cells (subset size $N_{\Omega_j}$) located at a distance $j \times 10$ μm ($j = \ldots -1, 0, 1, \ldots$) from the wave centre, and (ii) the local correlation of cell velocity:

$$\frac{1}{N} \sum_i \frac{1}{N_{\Phi_i}} \sum_{j \in \Phi_i} \hat{\boldsymbol{v}}_i \cdot \hat{\boldsymbol{v}}_j$$

where $\Phi_i$ denotes the subset cells (subset size $N_{\Phi_i}$) within a 50 μm radius of cell $i$ (*De Palo et al., 2017*). Local cell density was measured by the mean density of cells within a 60 μm radius of each cell.

## General theory of excitable media

This section outlines the elements of the theory we have used, and we used these to interpret spiral wave progression (*Tyson and Keener, 1988*; *Winfree, 2001*).

The eikonal relation dictates how the normal speed along the wave front $N$ changes due to wave curvature $K$:

$$N = N_0 + \varepsilon\, D\, K\,, \tag{1}$$

where $N_0$ is the planar (no curvature) wave speed, $D$ is the signal diffusion coefficient (which we assume remains constant), and $\varepsilon$ is the ratio of the rates of cell inactivation and activation (which we call the cell activation parameter; **Keener, 1986**). For a wave extending to the pivotal axis, the solution to **Equation 1** is accurately approximated by the following curve (in polar coordinates, $x = r\cos\theta\,(r,t)$ , $y = r\sin\theta\,(r,t)$):

$$\theta \approx \theta_0\,(t) + m\left(k\,(t)\,r + \log\left(\sqrt{3} + k\,(t)\ r\right)\right), \tag{2}$$

where $m$ is a constant. Parameter $k$ (the curve parameter) defines the curvature of the wave, and $\theta_0$ is the spiral angle of rotation, with rotation rate, $\omega = \dot{\theta}_0$ (**Tyson and Keener, 1988**). We fit **Equation 2** to the spatial distribution of maximally activated cells at each time point, throughout the entire progression of the spiral wave. The fit is accurate for all time points, which validates the use of the theory of excitable media even though cells are discretely distributed across space. This fit provides a measurement of the time evolution of the curve parameter $k$, rotation rate $\omega$, and hence the wave period $T = 2\,\pi/\omega$ (**Figure 4D**). The value of $k$ relates the planar wave speed to the cell excitability:

$$k = \frac{N_0}{\varepsilon\, D}, \tag{3}$$

Substituting **Equation 3** into **Equation 1** for $N_0$ allows the cell activation parameter $\varepsilon$ and planar wave speed $N_0$ to be expressed in terms of variables that can be directly measured from data:

$$\varepsilon = \frac{1}{D}\frac{N}{k+K}\,, \quad N_0 = \frac{N\,k}{k+K}\,. \tag{4}$$

Consistent with the traditional paradigm of spiral wave progression (**Höfer et al., 1995**; **van Oss et al., 1996**), we find that the cell activation parameter, which is the inverse of cell excitability, decreases during spiral wave progression (**Figure 5**). This result illustrates how we can identify key signalling properties of the cells from their population signalling patterns, at each time point during aggregation. Plotting $N_0$ from **Equation 4** against the wave period $T$ provides a prediction of the dispersion relation, which quantifies how a decrease in wave period reduces the wave speed because cells do not fully reset between waves, causing them to activate slower. The curvature relation describes the relationship between radius and rate of circulation of the spiral tip:

$$\varepsilon = \frac{\omega}{D\,\Omega\,(k, r_0)}\ , \quad \Omega = \frac{k\left(1 + 4kr_0 - \sqrt{1 + 8kr_0}\,\right)}{4\,r_0\,(1 + kr_0)}\,, \tag{5}$$

where $r_0$ is the radius of spiral tip circulation, which can measure from our data (**Tyson and Keener, 1988**). In support of our approach, the curvature and eikonal relations produce similar values of the cell activation parameter at all time points. We use the curvature relation to predict the circulation radius of the spiral tip based on our spiral measurements.

## Mathematical model (ABM)

A hybrid PDE ABM was developed to study the interplay between cell motion, spiral wave dynamics, and spatial patterning of cells around the spiral core.

External cAMP is secreted by active cells. We define active cells as having a positive activator value $A_i > 0$ (**Sgro et al., 2015**). Once secreted, we assume external cAMP diffuses and decays. The 2D external [cAMP] field $c$ was modelled by the following reaction-diffusion equation:

$$\partial_t c = D\nabla^2 c - \beta c + \sigma \sum_{i=1}^{N} H\left(A_i\right)\delta\left(\boldsymbol{x} - \boldsymbol{X}_i\right),$$

where $\beta$ and $\sigma$ are the rates of external cAMP decay and release from individual cells, $D$ is the external cAMP diffusion coefficient, $N$ is the number of cells, $H$ is the Heaviside function, and $\delta$ is the Dirac delta function, modelling the point source release of cAMP from cell $i$ ($1 \le i \le N$), with position $\boldsymbol{X}_i$ and activation state $A_i$. The spatial domain is $\Omega = \left[-\frac{L}{2}, \frac{L}{2}\right]^2 \subset R^2$ where $L > 0$ is the domain size. We used same parameter values of Vidal-Henriquez and Gholami (**Vidal-Henriquez and Gholami, 2019**).

The following modified FitzHugh-Nagumo model was used to describe the excitable release of cAMP from each cell:

$$\tau \, \dot{A}_i = A_i - \tfrac{1}{3} A_i^3 - I_i + \alpha \, \log \left(1 + \tfrac{c_i}{\kappa}\right) , \quad \tau \, \dot{I}_i = \epsilon \left(A_i - \gamma I_i + \mu\right),$$

where $I_i$ is the inhibitor level of cell $i$, and the value of each parameter was the same as *Sgro et al., 2015*, with exception of the time scale $\tau$ which was adjusted to match experimental data.

We tested several mathematical models of cell chemotaxis against the data for the velocity of individual cells as a function of the external cAMP field. The external cAMP field was inferred in our experimental data by substituting the measured position and discretised activation state of each cell in the reaction-diffusion equation above, which provides a prediction for the external [cAMP] at the location of each cell $c_i = c_i \left(X_i \left(t\right), t\right)$ and the spatial gradient $|c_i|$ (*Figure 7—videos 1 and 2*). The model that best fit the experimental data was a simple system of linear ordinary differential equations that encapsulate chemotactic memory:

$$\dot{\theta}_i = \alpha_m \, s_i \, \left|\nabla c_i\right| \, \sin\left(\theta_i^c - \theta_i\right) , \quad \dot{v}_i = \beta_m \, s_i \, \left|\nabla c_i\right| - \gamma_m \, v_i , \quad \dot{s}_i = \sigma_m \left(1 - s_i\right) - \lambda_m \, c_i \, s_i ,$$

where $0 \leq \theta_i < 2\pi$ and $v_i \geq 0$ are the direction and speed of cell movement, and $s_i \geq 0$ is the sensation variable to external cAMP as used in *Höfer et al., 1995*. This variable causes cell chemotaxis to be sensitive to the external cAMP gradient at the front of the wave, but not the rear. Variable $\left|\nabla c_i\right|$ and $0 \leq \theta_i^c < 2\pi$ are, respectively, the magnitude and direction of the external [cAMP] spatial gradient evaluated at $X_i$. Model parameters $\alpha_m$, $\beta_m$, $\gamma_m$, $\sigma_m$, and $\lambda_m$ were determined during the model identification using the Matlab function 'fminsearch' to obtain the best fit to experimental data. The first equation models the turning of the cell towards the local gradient of external cAMP. The turning speed depends on a cell's sensitivity and the size of the cAMP gradient. The second equation describes adaptation of the cell speed. It drives cell speed to be proportional to $s_i \left|\nabla c_i\right|$, i.e., higher sensitivity and larger local gradients will lead to faster speeds. Finally, the third equation models sensitivity: If a cell experiences a high concentration of external cAMP $c_i$, it will become de-sensitised, and it takes some time to become sensitive again. The variable $s_i$ provides 'inertia' or 'memory' that makes cell chemotaxis to be dependent on the history of external [cAMP] dynamics. When comparing the model predictions to the data, this effect was crucial to ensure that cells move towards the wave source at the wave front while also neglecting the wave rear. Specifically, between waves, the value of $s_i$ recovers to a state where the cells are sensitive to the external cAMP gradient at the front of the wave but subsequently $s_i$ reduces to a state where the cells are insensitive to the external cAMP gradient at the rear of the wave. For the final cell movement model, we also consider size exclusion effects between cells by including a cell-cell repulsion term. This means $\theta_i$ and $v_i$ are understood to mean the orientation and speed a cell would have if not in contact with other cells. We define as $d_{ij} = |X_i - X_j|$ the distance between the cell centres of the $i$-th and $j$-th cell. A cell's movement now follows:

$$\dot{X}_i = v_i \left(\cos\theta_i, \sin\theta_i\right) + \rho_R \sum_{j \neq i, \, d_{ij} < d_R} \left(1 - \tfrac{d_{ij}}{d_R}\right) \tfrac{X_i - X_j}{d_{ij}},$$

i.e., cells move with speed $v_i$ in direction $\theta_i$ in the absence of cell-cell repulsion. The second term describes a cell-cell repulsion with maximal repulsion magnitude $\rho_R$ where only cells within a distance of $d_R$ of each other exert a pushing force. See *Table 1* for parameter values.

We use no-flux boundary conditions for the external cAMP concentration, i.e., cAMP cannot leave the domain. Similarly, cells cannot leave the domain. Since the dynamics cause a concentration of cells towards the spiral centre, cells are being depleted from the very edge of the simulation domain, and hence, we only considered the middle part of the simulation domain to represent the biological situation. For all numerical experiments, we initialise cell positions $X_i$ and orientations $\theta_i$ using a uniform random distribution on $\Omega$ and $[0, 2\pi)$, respectively. Cell sensitivities $s_i$ were initially set to 1 and cell speeds $v_i$ to 0. The initial external cAMP concentration, activator, and inhibitor values were initialised as shown in *Figure 7—video 3*. The initial state was created by disabling cell movement and: (1) letting a vertical cAMP wave moving from left to right across a periodic domain until it is equilibrated, (2) setting the external cAMP in the upper-half of the domain to zero and setting the inhibitor value in the upper half on the domain to a high value (we used 2.5), and (3) setting the boundary conditions to no-flux and restarting the simulation.

**Table 1.** Model parameters.

| Symbol | Meaning | Value/unit | Reference |
|---|---|---|---|
| $c\left(\boldsymbol{x}, t\right)$ | External [cAMP] | Conc | |
| $\boldsymbol{X}_i\left(t\right)$ | Cell position | (mm, mm) | |
| $A_i\left(t\right)$ | cAMP activator | a.u. | |
| $I_i\left(t\right)$ | cAMP inhibitor | a.u. | |
| $\theta_i\left(t\right)$ | Cell orientation | Radians | |
| $V_i\left(t\right)$ | Cell speed | mm min$^{-1}$ | |
| $S_i\left(t\right)$ | Cell sensitivity | a.u., (0, 1) | |
| $\tau$ | Time scale of cAMP dynamics | 0.167 min | Fitted |
| $\varepsilon$ | Activator and inhibitor time scale | 0.1 | *Sgro et al., 2015* |
| $\gamma$ | ratio | 0.5 | *Sgro et al., 2015* |
| $\mu$ | Inhibitor decay rate | 1.2 | *Sgro et al., 2015* |
| $\alpha$ | Basal inhibitor production cAMP response | 0.058 | *Sgro et al., 2015* |
| $\kappa$ | magnitude cAMP response threshold | 10–5 conc | *Sgro et al., 2015* |
| $D$ | External cAMP diffusion constant | $2.4\times10^{-2}$ mm$^2$ min$^{-1}$ | (*Höfer et al., 1995*) Estimated |
| $\beta$ | External cAMP degradation rate | 5 min$^{-1}$ | |
| $\sigma$ | External cAMP secretion rate | $10^3$ conc mm$^2$ min$^{-1}$ | |
| $d$ | External cAMP secretion length scale | 0.1 mm | |
| $\alpha_m$ | Maximal turning rate | $3\times10^{-6}$ mm min$^{-1}$ conc$^{-1}$ | Fitted |
| $\beta_m$ | Speed model parameter 1 | $10^{-8}$ mm$^2$ min$^{-2}$ conc$^{-1}$ | Fitted |
| $\gamma_m$ | Speed model parameter 2 | 0.6 min$^{-1}$ | Fitted |
| $\sigma_m$ | Re-sensitisation rate | 150 min$^{-1}$ | Fitted |
| $\lambda_m$ | De-sensitisation rate | $4\times10^{-3}$ min$^{-1}$ conc$^{-1}$ | Fitted |
| $\rho_R$ | Maximum repulsion strength | 0.1 mm min$^{-1}$ | Estimated |
| $d_R$ | Cell radius | $5\times10$–3 mm | Measured |
| $L$ | Domain length | 4 mm | |
| $\Delta x$ | Spatial step | 0.02 mm | |
| $\Delta t$ | Temporal step | 0.05 min | |

Model equations were solved numerically using Matlab. The external [cAMP] field was solved across a 50 grid points per mm with the finite difference, implicit Euler method in time and central difference in space. Cell velocities and activation states were solved using the Matlab function 'ode45'.

## Mathematical model (PDE)

To test whether the coupling of cell excitability to the external [cAMP] dynamics sufficiently describes spiral wave progression, we extended a mathematical model of cAMP spiral waves to simulate the time evolution of the external [cAMP] field over time. We adopt the model by *Tyson and Murray,*

*1989*, which is of the general form of continuum excitable media with the dynamics of cAMP release governed by the Martiel-Goldbeter model:

$$\tau \frac{\partial u}{\partial t} = \tau \, \varepsilon \, D \, \nabla^2 u + \frac{1}{\varepsilon} \left( \alpha_u f_+ \left( u, v \right) - \beta_u u \right) , \quad \tau \frac{\partial v}{\partial t} = g_+ \left( u \right) \left( 1 - v \right) - g_- \left( u \right) v , \tag{6}$$

$$f_+ \left( u, v \right) = \frac{\lambda_1 + Y^2}{\lambda_2 + Y^2} , \quad Y = \frac{u \, v}{1 + u} , \quad g_+ = \frac{L_1 + \kappa L_2 c u}{1 + c u} , \quad g_- = \frac{1 + \kappa u}{1 + u}$$

where $u$ and $v$ (non-dimensional) are the spatial distribution of external [cAMP] (which can diffuse) and an internal inactivator (which cannot diffuse). We adopted the following non-dimensional parameter values from *Tyson and Murray, 1989*, most of which were rounded to the nearest order of magnitude: $\alpha_u = 10^3$, $\beta_u = 10^2$, $\kappa = 10^1$, $L_1 = 10^1$, $L_2 = 10^{-1}$, $\lambda_1 = 10^{-4}$, $\lambda_2 = 0.5$, $c = 10^1$. *Equation 6* is dimensionalised by the time scale $\tau = 0.075$ min and diffusion coefficient $D = 3 \times 10^4$ µm$^2$ min$^{-1}$. We note that solutions to this equation rapidly tend to an equilibrium state. Motivated by our finding that the cell excitability is coupled to the cAMP wave dynamics, we extended *Equation 6* by letting the cell activation variable, which we previously called the cell activation parameter (this inverse being a measure of cell excitability), to be a simple time-dependent function of the external [cAMP]:

$$\tau \frac{\partial \varepsilon}{\partial t} = \alpha_\varepsilon \, \left( \varepsilon_0 - \varepsilon \right) - \beta_\varepsilon \, u \, \varepsilon , \tag{7}$$

Where $\alpha_\varepsilon = \frac{1}{50}$, $\beta_\varepsilon = \frac{35}{50}$, $\varepsilon_0 = 10$. *Equation 7* states that the cell activation variable tends to a baseline value $\varepsilon_0$ and decreases in response to external cAMP. The simulations were performed on a 4.5 mm × 4.5 mm grid with 0.025 mm spacing and 0.05 min time step. The initial condition was the state where $= 0$, $v = 0$ , $\varepsilon = \frac{\varepsilon_0}{2}$, except for a rectangular region representing a broken wave, with $u = 10$ and $v = 0$ and to one side $u = 0$ and $v = 1$. To numerically solve *Equations 6; 7*, we used the implicit finite difference method for the diffusion term and the Runge-Kutta (Matlab 'ode45' function) method for the reaction terms. The Neumman boundary condition (no flux) was used.

## Acknowledgements

This study was supported by a Wellcome Trust Senior Fellowship (202867/Z/16/Z) to JRC. The funder had no role in study design, data collection and interpretation, or the decision to submit the work for publication. We thank Elizabeth Westbrook, Ricardo Barrientos, Courtney Lancaster, Kees Weijer, and Allyson Sgro for comments on the manuscript.

## Additional information

### Funding

| Funder | Grant reference number | Author |
| --- | --- | --- |
| Wellcome Trust | 202867/Z/16/Z | Jonathan R Chubb |

The funders had no role in study design, data collection and interpretation, or the decision to submit the work for publication. For the purpose of Open Access, the authors have applied a CC BY public copyright license to any Author Accepted Manuscript version arising from this submission.

### Author contributions

Hugh Z Ford, Conceptualization, Data curation, Software, Formal analysis, Validation, Investigation, Visualization, Methodology, Writing - original draft, Writing - review and editing, Mathematical modelling (PDE), Molecular biology and cell line generation; Angelika Manhart, Software, Formal analysis, Mathematical modelling (ABM); Jonathan R Chubb, Conceptualization, Resources, Supervision, Funding acquisition, Writing - original draft, Project administration, Writing - review and editing, Molecular biology and cell line generation

### Author ORCIDs

Hugh Z Ford http://orcid.org/0000-0002-0457-7224
Jonathan R Chubb http://orcid.org/0000-0001-6898-9765

Decision letter and Author response
Decision letter https://doi.org/10.7554/eLife.83796.sa1
Author response https://doi.org/10.7554/eLife.83796.sa2

## Additional files

### Supplementary files
- MDAR checklist
- Source code 1. Matlab code for agent-based model of cell aggregation.

### Data availability
The high resolution movies of cAMP signalling and optogenetic treatments can be accessed at UCL's institutional repository using the DOI: 10.5522/04/21360975. All image analysis and mathematical modelling methodology described in full in methods section. Matlab code for the agent-based model is provided.

The following dataset was generated:

| Author(s) | Year | Dataset title | Dataset URL | Database and Identifier |
|---|---|---|---|---|
| Chubb J, Ford H | 2022 | cAMP signalling movies and optogenetics | https://doi.org/10.5522/04/21360975.v1 | figshare, 10.5522/04/21360975.v1 |

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
