## [Editor Report]

This fundamental work substantially advances our understanding of how multicellular structures transmit information over long ranges. Compelling approaches combining experiments and theory unravel the mechanism by which amoeba form migrating cellular waves by chemotaxis. The work will be of broad interest to cell and developmental biologists.

---

## [Decision Letter]

**Decision letter after peer review:**

[Editors’ note: the authors submitted for reconsideration following the decision after peer review. What follows is the decision letter after the first round of review.]

Thank you for submitting the paper "Structural plasticity of signalling centres enables long-range frequency modulation of morphogenesis" for consideration by *eLife*. Your article has been reviewed by 2 peer reviewers, and the evaluation has been overseen by a Reviewing Editor and a Senior Editor. The following individuals involved in the review of your submission have agreed to reveal their identity: Vincent Calvez (Reviewer #1); Satoshi Sawai (Reviewer #2).

Comments to the Authors:

We are sorry to say that, after consultation with the reviewers, we have decided that this work will not be considered further for publication by *eLife*.

The reviewers commend the authors for developing impressive cellular tracking to link *Dictyostelium* single-cell behaviors to mesoscale patterns for the first time. However, while the data is impressive, many of the data analyses were confirmatory of previous more macroscopic observations, and the reviewers could not identify clear improvements in the biological hypotheses nor in the conclusions from the model, in comparison with previous modeling contributions (see e.g. Höfer et al. and further references in the attached reviews).

While we cannot publish the present manuscript, we nevertheless see the potential of this approach and would be willing to consider a re-submission provided that the new experimental and theoretical analyses provide important new concepts to current models.

The individual reviewers' reports appended below might help to orient the work in this direction. Note however that at this stage, we cannot guarantee that the new manuscript will be re-reviewed and if so, it will have to go through another full round of review.

*Reviewer #1 (Recommendations for the authors):*

The complex transition from unicellular to the multicellular stage in Dicty is described in an accurate way. The authors focus on the dynamics of the signaling centers which are the core of the spiral cAMP waves. The waves propagate in a homogeneous, then in a heterogeneous population, where the cells act as relaying the signal (cAMP release when activated), similarly to the propagation of Mexican waves. The propagation of such waves in excitable media has been studied thoroughly in the literature, and in particular in the context of Dicty development. However, several questionable hypotheses were usually set in order to account for the key dynamical features of the waves (e.g. increase in frequency and decrease in wave speed). For instance, the authors point to a celebrated work by Höfer et al. (1995) where it is assumed that cAMP excitable dynamics are density-dependent, in contrast with recent investigations.

My understanding is that the authors come up with an explanation of the aggregation dynamics based on first principles involving the geometry of the singularity vortex at the core of the spiral waves, coupled with directed cell motion (chemotaxis). Indeed, they describe the motion of the wave tip as being essentially circular (ring). The nature of this ring is both "molecular" (the trajectory of the cAMP wave tip), and "cellular" (as cells organize themselves as a dense ring of cells). This core ring plays the role of a pacemaker, although it seems to be an emerging property of the system. It is shown to contract slowly in time, in agreement with previous studies. By tracking the cells around the ring, the authors could show that cells are moving in the opposite direction of the wave tip, with a net tendency to move inwards, resulting in the contraction of the ring. This, in turn, makes the wavelength decrease (increase in frequency) and the wave speed decrease as well. This change in the dynamical properties of the wave at a very fine scale (few cell diameters) has a long-range influence, as it is observed that cells organize in streams towards the signaling center only beyond a certain threshold in frequency (around 1/4.5 min), for which cell persistence overcomes random motion.

The authors also compile previous modeling contributions in order to build a hybrid model that reproduces the experimental observations in a very satisfactory way. The model focuses on a specific stage of the developmental process (after the onset of spiral waves, before the formation of multi-arm waves). The model consists of a "molecular" part (continuous) based on recent progress (including Fitzhugh-Nagumo dynamics), and a "cellular" part (discrete). Interestingly, the parameters of the molecular side of the model have been taken from literature, with the exception of one timescale which has been fitted to the data. In contrast, the cellular component has been subject to model selection, resulting in cell motion with inertia.

The strength of this work is certainly the comprehensive description of a complex developmental transition, informed by basic geometrical, molecular, and cellular principles (spiral wave propagation, signal relay, chemotaxis). I appreciate the fact that the dynamics are described at the local scale (the spiral vortex), and at the population scale (cellular streams). I also appreciate that data have both a molecular and a cellular nature (cAMP reporter and cell tracks, respectively), allowing for multiple views on the pattern, together with the design of a very convincing mathematical model. One of the major deductions of this work is that the modulation in frequency (the global developmental signal) is due to cell reorganization by chemotaxis at a fine scale (the vortex ring), rather than a change in the excitability properties of the system.

As for weaknesses, I would pinpoint the fact that the mathematical model is meant to describe the experimental observations. This is done in an admirable way. However, I would have appreciated a critical discussion of the modeling assumptions and their consequences. More precisely, I suspect that the details of the chemotactic rules are essential to observing the correct pattern (both the ring contraction and the cellular streaming). However, the details of the model selection are not given, so it is difficult to evaluate whether each component of the sophisticated chemotactic rule is essential. Alternatively speaking, what is the role of inertia/encoding of memory effects in the emergence of the macroscopic pattern?

A minor criticism is about the transition to the three-arms spiral, which is described without modeling support (although the model is seemingly in place). This discrepancy between a thorough investigation of the dynamics of the single spiral, and the rather quick description of the transition from one to three arms looks unsatisfactory at first glance.

In my opinion, this work is a very impressive combination of experimental data and modeling study. The transition at play is very well documented in the literature (dating back to the '70s), and so valuable contributions must bring clear progress. I think it is the case here, with a subtle interplay between local and global scale, and with a clear view of both the cells and their environment.

1. I would like to see more discussion about the criticism of what is presented as a key hypothesis in Höfer et al. (1995).

I quote: "A previous mathematical model (Hofer, Sherratt and Maini, 1995), using the Martiel-Goldbeter model of cAMP release (Martiel and Goldbeter, 1987), suggested that this contraction is due to an increase in excitability due to an increase in cell density around the ring."

I looked into the original paper, and I found that the local increase in excitability is due to the density-dependence of the cAMP dynamics (Eq. (4) in Höfer et al., in particular, the production rate \λ.phi(n)). In the model presented here, there is also a density-dependence in the cAMP production rate (Line 628), as each cell body acts as a source term for cAMP (when activated). So, I do not really see the conceptual difference between the model here and the one in Höfer et al. in terms of cAMP dynamics. Please elaborate.

Ideally, the model would enable disentangling these effects (potential increase in excitability, and spatial reorganization) towards a stronger conclusion.

In fact, quoting "However, spiral tip circulation holds a fixed radius in simulations without cell motion, suggesting that the change in cell spatial distribution is required to explain the contraction in tip circulation", it seems to me that the same explanation holds in Höfer et al., where the local increase in density gradually increases the wave frequency by sustained excitability.

Further in the manuscript, quoting "We deduce that it is the rearrangement of cells via chemotaxis, not the increase in cell excitability, that drives vortex contraction." That is precisely my point: as far as I understand, in Höfer et al. the rearrangement induces higher excitability, and formally I noticed similarities in the modeling of cAMP dynamics.

A related observation follows from the quotation "We note that other, more complicated dynamics of external cAMP have been used (e.g. active degradation by cells Hofer, Sherratt and Maini, 1995), however here we chose to omit such potential effects in the interest of simplicity."

I think it is worth checking that it does not change the dynamics, even if cells actively degrade/consume cAMP.

2. The chemotaxis models differ significantly from classical (basic) models, for it includes inertia. I understand that the authors follow a data-driven approach, extending the standard model to better match the observed trajectories in a sampled area. Then, it is very satisfactory to reproduce the full macroscopic picture.

I suspect that inertia plays an important role, as it encodes memory effects in the cell response to the signaling gradients. Would it be possible to emphasize the important features of the chemotactic model? Practically, would it be possible to link some of the chemotactic parameters to the frequency threshold beyond which the cells follow streams (Figure 2)?

Similarly, it would be highly valuable, I think, to compare the results in Figure 4D with the model.

*Reviewer #2 (Recommendations for the authors):*

The study by Ford et al. entitled "Structural plasticity of signalling centers enables long-range frequency modulation of morphogenesis" reports on the multi-scale live-cell imaging analysis of cAMP waves and aggregation of *Dictyostelium* discoideum cells. By simultaneously imaging low-magnification images at millimeter-scale and high-magnification images at micrometer-scale the authors attempted to obtain full-resolution data at the single-cell level of an entire population-level dynamics. The wavelength of the cAMP waves encompasses close to a hundred micrometer and the entire aggregate will be close to a millimeter. Thus traditional imaging study utilized dark-field macroscopy which highlights the cringing cell shape change that highly correlates with the cAMP changes (Tomchik & Devreotes, Science 212, 443-, 1981) which were not compatible with single-cell resolution observation. Because the waves are self-organizing emergent dynamics, where it is initiated and how it evolves in terms of their frequency and geometry depends highly on the stochasticity and variability of the cAMP relay response of individual cells and how extracellular phosphodiesterase which degrades extracellular cAMP is distributed in space. An earlier study using a FRET-based cAMP sensor and perfusion experiments has shown that below a nanomolar cAMP, individual cells exhibit a random transient synthesis of cAMP(Gregor et al. Science 328, 1021-, 2010). Due to the low signal-to-noise ratio of the FRET imaging, the study only showed a spatial correlation of the firing events and not single-cell level data. From this and other observations regarding how cAMP is amplified and what dictates its responsiveness, it has been thought that the system evolves in sequence from stochastic firing to collective firing then to repetition of this synchronized event (i.e. oscillations)(Gregor et al. Science 328, 1021-, 2010). With single-wave length cAMP sensor Flamindo2 which has much higher s/n than the FRET sensor, this work for the first time clearly shows this sequence event at the single-cell level marking it the first of a kind recording self-organizing multicellular organization. The data show that before the signaling becomes fully synchronized in the population the distribution of inter-firing intervals is multi-peaked. I congratulate the authors for the clear demonstration.

Weakness:

The work did not address the consequence of cell-cell heterogeneity with regard to the wave initiation and spiral formation which could take further advantage of their high-resolution high-sensitivity imaging. In an earlier study, cells that were fully developed in shaken culture are shown to develop highly synchronized waves and no spirals (Sawai, Thomason, Cox, Nature 433, 323-, 2005). The initiation of the spiral is inherently linked to cell-cell heterogeneity in the responsiveness and refractoriness as the authors mentioned. Since this has not been shown explicitly at the single-cell level, a detailed analysis of this event would have added novelty to the work substantially. The curvature vs periodicity relationship has already been reported earlier based on the dark-field optics (Foerster Muller Hess, Development 109, 1-16, 1990), and thus this by itself is not new. The authors also studied a mathematical model and proposed that shrinking cellular ring at the spiral core requires cell migration. Their interpretation of how this occurs, however, is not satisfactorily vindicated experimentally and is somewhat counter to an earlier report on caffeine-treated cells that fail to shrink this ring despite they still migrate (Siegert and Weijer, J Cell Sci. 93, 325-335, 1989). These earlier experiments and previous modeling studies that consistently explain this from the change in local excitability and cell density were not revisited here to support their alternative scenario.

The authors chose to center the thesis around testing the frequency selectivity of collective cell migration. While the present results are important and do open an avenue for future studies as to how cell chemotaxis and migration can be encoded by extracellular signals, this part of the thesis is not vindicated experimentally and is just a suggestion as the authors also described. It has been known that the start of high-speed cell migration coincides with the timing when the cAMP oscillation switches periodicity from 5 to 3 minutes (Rietdorf , Siegert, Weijer Dev. Biol. 177 (Figure 1A, C), 427-, 1996, Dormann et al. Curr Biol. 12, 1178-, 2002, Hashimura et al., Commun. Biol. 2, 34, 2019). So the observation is not entirely new. Also, because imaging alone will only show correlation and not causality, their main thesis regarding the dependence of cell migration on the cAMP frequency remains to be tested experimentally by input/output analysis at the single-cell level. Recent such analyses make use of a perfusion chamber to generate artificial waves (Skoge et al., PNAS 111, 144480, 2014, Nakajima et al., Nat. Commun. 5, 5367, 2014, Nakjima et al., Lab Chip 16,4382, 2016) suggest other possibilities such as dependence on history of repetition which serves as a developmental counter – and/or the causality can be reverse i.e. the periodicity may change because cell takes highly polarized form. Artificial wave experiments have demonstrated that not 1 to 2 but 4-5 passages of 5.5-min periodicity waves can induce cell polarity or the memory effect where cells become elongated and do not stop at the wave back. This points to a "counting" mechanism, not frequency, thus it is unclear whether a 3-min frequency-specific collective migratory response is present or necessary for the onset of high motility migration as authors seem to suggest.

This was a very enjoyable piece of work and I congratulate the authors on this achievement. Scientifically, however, I see several major caveats that I hope the authors can address when revising the manuscript.

Figure 1 shows imaging results of the onset of collective signaling. This is a great refinement of the work by Gregor et al. (Science 2010) where they have looked at small populations of cAMP FRET sensor expressing cells and analyzed firing events and different spatial locations. Whether random firing occurs prior to synchronous firing was not clear in Gregor et al. but was inferred from stochastic firing response when cells were stimulated at sub-nanomolar extracellular cAMP using a perfusion chamber. The present work provides much stronger evidence of this sequence of events proposed by Gregor et al. where the accumulation of stochastic firing at the single-cell level leads to synchronous firing. The FRET measurements by Gregor lacked the necessary S/N ratio and such fine resolution dynamics at the population level at the time were hard to come by. The authors' use of Flamindo is thus a very nice approach. However, again authors failed to mention that the use of Flamindo to analyze wave dynamics is not without precedent (Hashimura et al., Communications Biology 2, 34, 2019). This paper should be cited.

Figure 2 addresses the link between the wave and cell migration. Here and elsewhere in the text, I am afraid that the present manuscript uses the term 'information' quite loosely. It is worth noting that the information processing property of this system has been analyzed quite extensively, and this could have been elaborated better in the text with a more balanced view of various possibilities. The current observations should be discussed in light of these single-cell level input-output relations. The fact that frequency increases at the timing of streaming and the timing of spiral evolution has been shown by Antony Durston ("The control of morphogenesis in *Dictyostelium* discoideum" in Eucaryotic Microbes as Model Developmental systems 1977) and more accurately by others as described in the public review. Some novelty I see in Figure 2 data is the quantitative analysis of the spiral core size and change in the frequency. However, from simple observations, as shown in the present paper, it is not possible to decipher causality from correlated events. The caveat is that one cannot discriminate the effect of the number of waves experienced by the cell versus the frequency, periodicity versus wave speed. In the text, the former is completely ignored or maybe they are mixed up. I don't see clear evidence of frequency modulation. Early works to tackle this challenge were pioneered by Michael Vicker and then by David Soll and his colleagues that showed with the use of perfusion setup that cells read out temporal information rather than the spatial gradient. This aspect was further vindicated by Nakajima et al. (Nat. Commun. 5, 5367, 2014) that employed an isolated single-passage of artificially generated wave that showed that it is the wave passage time (wave speed) that dictates what is being read out. The current understanding derived from this study and Skoge et al. (PNAS ) is that the adaptive response by the Local Excitation Global Inhibition (LEGI) framework (Levchenko & Iglesias) is fully capable of describing the readout property. At a slow wave speed, cells are provided enough time to readout spatial gradient at the steady-state response. At fast waves, the directionality is first determined by the first hit mechanism which is based on the delay of stimulus arrival and the following transduction across the cell. This makes use of transient response and requires temporally increasing extracellular cAMP thus occurring specifically at the wavefront and not waveback. Much faster wave speed effectively makes the stimulus spatially uniform not allowing cells to form a localized leading edge. What does require an extension to the LEGI framework is this fast-wave regime. At the 3-min interval stimulus, the system comes somewhat close to the limit capable to make use of the first hit mechanism. One possible resolution to this paradox is the hysteresis/memory effect. The authors cite Skoge et al., which is very important here for data interpretation, but there is another work as relevant that employed artificial rotating waves to induce circular cell migration (Nakajima et al., Lab Chip 22, 4382, 2016; Figure 6f). There, 4-5 passages of 5.5-min periodicity waves can induce cell polarity or the memory effect where cells become elongated and do not stop at the wave back. Given this prior observation, I just don't see the need for a 3-min frequency-specific collective migratory response as the authors seem to suggest. Given intermediate wave speed stimulus first, cells will obtain this highly polarized state and thus could move by "steering" mode of chemotaxis around the spiral core. It could be that 3.5 min periodicity is a consequence of cells taking high polarity – in this case, the causality is reversed. I believe the manuscript would be greatly improved if suggest authors take these possible scenarios into consideration in the Discussion section.

Figure 3. The fact that cells move towards the incoming spiral waves is well studied and established. I am not sure what the authors wanted to test here (L. 206-209).

Figure 4 addresses the curvature and the dispersion relation. The quantification of the dispersion relation at this resolution is certainly nice, however, the authors failed to mention that the curvature relation has been characterized earlier by Foerster Muller Hess "Curvature and spiral geometry … (Development 109, 11-16, Figure 6 for the spiral geometry fitting and Figure 3,4 for the dispersion curve). This paper is cited in this manuscript but only in general terms and not this specific point which was essentially the central theme of the Foerster paper. I could well be missing something, but since the authors have not put the results in the light of the earlier finding, the importance of the present data is hard to grasp. In the figure legend, it refers to ref. [41] for the parameter epsilon, however, the reference is not numbered.

Figure 5 and L. 300-306 describe model simulations and the cell density effect. By comparing model simulations with and without cell motion (Figure 5D and 5E), the authors suggest that the change in cell distribution is required to explain the contraction in tip circulation. They suggest a scheme where the cellular ring constrains spiral tip motion and the extracellular cAMP field guides cells more inwards. How this scheme differs from those proposed by others (Hoefer and Maini) is not straightforward. This is hard to test experimentally and may be ill-posed. Making cells immobile by F-actin destabilizing drug and substrate adhesion also affects the cAMP synthesis (Interface 13 20160233, 2016), and thus cell motility and the cAMP-relay is highly intertwined and the parameters are mutually dependent. Cells treated with caffeine which inhibits cAMP synthesis in *Dictyostelium* do show open rings that fail to shrink and the core size depends on caffeine concentration (Siegert and Weijer JCS 93, 325-335, 1989). There, the cells are moving but the ring fails to contract and thus contradicts the authors' present thesis. What is missing here is an experimental demonstration of the side effects of caffeine that affect chemotaxis and cell motility. It would be very interesting to apply the current analysis to caffeine-treated cells. For example, caffeine affects the TORC2 pathway. But again, this is very hard to test as both chemotaxis and the cAMP-relay share a common signaling pathway.

Sgro 2015 and Kamino 2017 (L. 304) do examine cell-density independence of oscillatory instability arising from fold-change response characteristics. It addresses important but rather idealized general conditions necessary for robustness to cell density fluctuations and thus it may not be adequate for the more system-specific detailed analysis required here. The abstract nature stems from the fact that density dependence comes mainly at the level of cAMP detection and synthesis. In reality, how extracellular PDE production is regulated most likely changes this cell density dependence drastically, and thus the use of these models here does not warrant a meaningful conjecture. cAMP wave geometries do depend strongly on cell density as has been demonstrated by Lee, Cox, and Goldstein (Phys Rev Lett 76, 1996). The Martriel-Goldbeter model is not just a model of cAMP release (L.301). It describes cAMP synthesis, cAMP degradation, and receptor adaptation. It is still the most consistent model with the biochemistry bulk-based analysis by Peter Devreotes and single-cell-based cAMP-relay analysis as described below. The direct comparison of the current simulations should also be made with the partial differential equation-agent hybrid-based simulations by van Oss et al. Weijer (J theor Biol 181, 203-213, 1996) which are very similar in approach and methodology except that it is based on Martiel-Goldbeter model. This important work is not discussed nor cited. They have made a good analysis of the relationship between the wave speed, cell movement, and cell density around the spiral core (Figure 3 of this reference). Wave speed increases with increasing cell density and the core shrinks.

Figure 7 addresses cell-cell variability during spiral development. The collective cAMP oscillations/waves in *Dictyostelium* require a receptor-mediated extracellular feedback loop both for amplification (via cAMP) and also for de-adaptation (clearing of extracellular cAMP by extracellular PDE). Dependence on the cAMP receptors is well-addressed by various hybrid receptors and mutations (Dormann et al. JCS 114, 2513-, 2001). Temperature shift mutation of ACA can turn waves on and off (Patel et al. EMBO 19, 2247-,2000). Oscillations can also be turned on or off in the PDE-null mutant by mimicking degradation by assisted dilution by perfusion (Masaki et al., Biophys 104, 1191-, 2013). The major unresolved question regarding the system is how far we understand these properties at the single-cell level. There are some cell-cell statistics in this work but they can be highlighted. This is important as how waves are initiated and how spirals develop do depend heavily on cell-cell variability in the related parameters along the developmental path (Lauzeral & Goldbeter PNAS, Levine et al. PNAS 1996, Sawai, Thomason & Cox Nature 2005). At the single-cell level, the average response characteristics are well characterized (Gregor et al. Science 2010, Kamino et al. PNAS). The response dynamic as well as the fold-change conditions necessary for the amplification is consistent with the macroscopic behavior of the system. As for the refractory period, dependence on ligand clearing (which is what PDE does) has been quantified at the single-cell level (Fukujin et al., Interface 13 20160233, 2016). Here it is shown that 10 min of ligand clearing is required for full response recovery while at 3.5 min washing, the response is attenuated to about 50% in amplitude. The absence of the absolute refractory period and the fact that cells can still respond after 3.5 min ligand clearing should be noted and discussed (L. 354- and elsewhere). Also, according to Gregor et al. repetitive stimulus at 3 min intervals will eventually attenuate the cAMP-relay response. Could this be related to the final extinction of the waves? All these important points are highly relevant when the analysis is zoomed in at the single-cell level resolution.

[Editors’ note: further revisions were suggested prior to acceptance, as described below.]

Thank you for resubmitting your work entitled "Controlling periodic long-range signalling to drive a morphogenetic transition" for further consideration by *eLife*. Your revised article has been evaluated by Naama Barkai (Senior Editor) and a Reviewing Editor.

The manuscript has been improved but there are some remaining issues that need to be addressed, as outlined below:

The reviewers have discussed their reviews with one another, and the Reviewing Editor has drafted this to help you prepare a revised submission. While both reviewers appreciate the improvement and I depth modifications made to this version of the manuscript they still raise significant concerns that should be addressed prior to publication.

Clarifications must be added to explain the modeling:

The authors draw three hypotheses:

1. A "global" scenario, where the overall signaling dynamics are coupled with cell density, essentially following Hofer et al. Thus, cell reorganization through chemotaxis can drive changes in the excitability properties of the (inhomogeneous) medium, with a significant increase at the center when cells aggregate. This scenario is ruled out based on a predicted temporality (cell massive reorganization precedes changes in wave dynamics) which is in contrast with the experimental observations.

2. A "local" scenario, where the local reorganization of cells near the tip dramatically reduces both its radius and its speed (but not its rotation period, which is a drawback). The conceptual difference with the model of scenario 1 is not obvious. The models are of a different nature (continuous field versus the individual description of the cell population). Nevertheless, the release of external cAMP (bottom of p.33) increases with the local cell density, which could be put in similarity to the work by Hofer et al. In the latter model, the cell density-dependent "excitability" is motivated as follows: "An increase in cell density increases the local production and degradation rates per volume element, and it decreases locally the share of the extracellular volume in a volume element."

Thus, it is not justified as an increase of "individual excitability", as the authors describe it here on p.14: "In this view, the cell excitability increases by virtue of cells accumulating at the spiral core through chemotaxis". In fact, in Hofer et al., the increase in cell density changes the excitability locally in the medium, but this might not be interpreted as an increase in individual cell excitability (in contrast with Hypothesis #3).

3. A "molecular-only scenario", with no coupling with the cell density. The idea is that the cells modify their individual excitability state when waves are passing through. The additional equation taking into account this new idea is minimal, which is nice. The cell re-organization follows from chemotaxis, but there is no feedback in the model (actually, the cell density is not even described, see Figure 8).

Comments (towards a clarification of the three hypotheses):

Overall, the modelling part is convincing. It could be interesting to present the two alternatives (providing that Hyp #1 and Hyp #2 are not conceptually different, see below) as follows:

"emerging collective excitability by cell reorganization with constant individual excitability" (#1 and #2) versus "adaptative individual excitability in a homogeneous cell density".

Is that (nearly) correct?

The agent-based model of Hypothesis #2 may not be equivalent to the Hofer et al. model, but what are the exact conceptual differences between them? Along this line, the results of Hypothesis #2 may have similar inconsistencies with the observations in Hypothesis #1. Indeed, it looks like in Figure 7, the cell density in the model shows dramatic changes after 1h. The authors argued this was a clear way to remove Hypothesis #1. Isn't it the case here also?

This comment is of little importance for the conclusion of the paper, as both Hypotheses #1 and #2 are ruled out after all. But this is a source of confusion.

Another source of confusion was the apparent arbitrariness in the choice of the molecular "excitable" systems in the two hypotheses 2 and 3. The Martiel-Goldbeter (MG) system is used to test hypothesis 2, whereas, the more basic Fitzhugh-Nagumo (FHN) system is used to test Hypothesis 3. This may be not an important drawback, since the two models belong to the same class of excitable systems. However, this may add some confusion and some possible weakness to the overall conclusions.

It is understood that the authors followed a recent modelling approach for testing hypothesis 2 (although the cAMP internal dynamics are somewhat basic equations of FHN type, as compared to the MG model). However, they used some modified version of an older series of work from the late 1980s to test hypothesis 3.

Moreover, the formulation of the model in hypothesis 2 is ambiguous:

"We adopt the model by Tyson et al. (Tyson and Murray, 1989), which is of the general form of continuum excitable media (equation (1)) with the dynamics of cAMP release (functions f and g in equation (1)) governed by the Martiel-Goldbeter model…"

However, there is nothing like "f and g" in equation (1), and moreover, the MG system (6) is a closed one, and equation (1) should be viewed as an asymptotic limit capturing the geometric features of the system (6).

Also, references are missing in the section "General theory of excitable media" on page 32. I understood that the main reference is Tyson-Keener (1988), but the equation (1) as written in the present ms, is seemingly written as N = N0 + DK [Tyson-Keener 1988, page 10, left column]. The additional parameter epsilon in (1) is dimensionless, so it would make perfect sense if supported by a suitable reference (and it has a clear biological interpretation).

The authors also don't describe how this excitability change comes about – likely the level of extracellular cAMP. The oscillations and waves in this system may be taking place at the nanomolar range at the basal or several hundred nanomolar ranges at the basal as long as there is enough fold-change increase. One would assume that higher cell density is necessary to support a high basal level state however if the period decreases degradation of cAMP fails to catch up with the frequency. This cannot be read from the intracellular cAMP indicator and is important to interpret from the model.

*Reviewer #1 (Recommendations for the authors):*

I consider this manuscript as a new submission since the conclusions have been significantly changed. I found the new version highly valuable, with a new hypothesis that is conceptually different than the previous ones and explains most of the observations. Moreover, this new hypothesis has been tested with an original experiment, and it succeeded where the previous paradigm failed.

Main comment:

The authors draw three hypotheses:

1. A "global" scenario, where the overall signalling dynamics are coupled with cell density. Thus, cell reorganization through chemotaxis can drive changes in the excitability properties of the (inhomogeneous) medium, with a significant increase at the centre when cells aggregate.

This scenario is ruled out based on a predicted temporality (cell massive reorganization precedes changes in wave dynamics) which is in contrast with the experimental observations.

2. A "local" scenario, where the local reorganization of cells near the tip dramatically reduces both its radius and its speed (but not its rotation period, which is a drawback). I'm afraid I don't quite see the conceptual difference with the model of scenario 1. The models are of a different nature (continuous field versus the individual description of the cell population). Nevertheless, the release of external cAMP (bottom of p.33) increases with the local cell density, which could be put in similarity to the work by Hofer et al. In the latter model, the cell density-dependent "excitability" is motivated as follows: "An increase in cell density increases the local production and degradation rates per volume element, and it decreases locally the share of the extracellular volume in a volume element."

Thus, it is not justified as an increase of "individual excitability", as the authors describe it here on p.14: "In this view, the cell excitability increases by virtue of cells accumulating at the spiral core through chemotaxis".

I would put it like this: in Hofer et al., the increase in cell density changes the excitability locally in the medium, but this might not be interpreted as an increase in individual cell excitability (in contrast with Hypothesis #3).

I don't claim that the agent-based model of Hypothesis #2 is equivalent to the Hofer et al. model, but I don't see conceptual differences between them. In this line, I was asking myself if the results of Hypothesis #2 do not suffer the same inconsistency with observations as of Hypothesis #1. Indeed, it looks to me that in Figure 7, the cell density in the model shows dramatic changes after 1h. The authors argued this was a clear way to remove Hypothesis #1. Isn't it the case here also?

This comment is of little importance for the conclusion of the paper, as both Hypotheses #1 and #2 are ruled out after all. But this was a source of confusion for me.

3. A "molecular-only scenario", with no coupling with the cell density. The idea is that the cells modify their individual excitability state when waves are passing through. The additional equation taking into account this new idea is minimal, I appreciate it. The cell re-organization follows by chemotaxis, but there is no feedback in the model (actually, the cell density is not even described, see Figure 8).

I found the modelling part very convincing. I may suggest presenting the two alternatives (providing that Hyp #1 and Hyp #2 are not conceptually different, but this is my own understanding) as follows:

"emerging collective excitability by cell reorganization with constant individual excitability" (#1 and #2) versus "adaptative individual excitability in a homogeneous cell density".

Is that (nearly) correct?

Side comments:

"Altogether, our results describe how the collective behaviour of 10^5 -10^6 cells across several millimetres is organised by the signalling dynamics of a few hundred cells."

– I'm not sure I understand what is meant here. I agree that the whole organization relies on a "pacemaker", that may be constituted of a few hundred cells (although I don't know what supports this number, is there a spatial threshold to decide where is the pacemaker?). In particular, as far as I understand the model, the cells are all equally capable of producing and relaying the signal in the same terms. It is thus the geometry of the process, and not the cell heterogeneity (leaders vs. followers) which drive the dynamics.

I would like to make it clear that "the few hundred cells" are not considered as different from the other in terms of the capability of making the pattern (would there be some artificial exchanges within the population)? But maybe I got it wrong.

Figure 7/8 (titles) – They are both mathematical models, so the titles should be updated (in reference to the corresponding hypotheses 2/3 maybe?).

Figure 4-D-1st and Figure 6-B-2nd – I understand that this is the same figure. Is that correct? (a few points are different, though).

In the methods, it seems that the models which are used in the simulations are presented in the reversed order (Hypothesis #3, then Hypothesis #2). Is there a good reason for this?

*Reviewer #2 (Recommendations for the authors):*

Authors: "Thanks for pointing out these issues. Importantly, we have now carried out a substantial series of experimental perturbations of wave frequency, using optogenetic activation of the bPAC adenylyl cyclase. Specifically, we generated trains of pulses of different fixed frequencies and also, trains of gradually increasing frequency. The data (presented in the new Figure 3) argue strongly for migration specified by the frequency by which waves are generated by the signalling centre. Additional data corroborating these effects in the unperturbed state are shown in Figure 6B. In particular, we also show that many facets of the wave properties (width, speed) are determined by the frequency at which waves are generated from the signalling centre (Figure 5B). As the Reviewer suggested, we have generalised these new results in the discussion, in the context of the diverse literature the Reviewer highlights. "

Reviewer response – The authors have shown that 3 min periodic forcing gives rise to conglomerates of many aggregates which they interpret as backing their thesis that fast frequency supports the collective motion. The other likely possibility is that 3 min periodic forcing entrains other wave territories and this demonstration while it is very nice does not prove by itself that fast frequency is necessary for the transition to collective motion.

Authors: "Hofer model and the cell patterning model. Each model has its own subsection in a section entitled "Hypotheses". We have also proposed an additional hypothesis, that contrasts both models.

Regarding, experimental testing, we have incorporated new data where we explicitly evaluated the Hofer model (cell density). These data are shown in Figure 6. Specifically, we see no strong coupling between cell density and either excitability or wave period. Spiral wave progression proceeds (and is largely completed) prior to changes in density.

To test the ring closure hypothesis, while avoiding the caveats of drug treatments, we devised a mechanical perturbation experiment, in which we block the hole closure using an agarose block (Figure 9). This experimental test revised our thinking of the mechanism. We strongly thank the reviewer for encouraging us to think about this."

Reviewer response – The new data is quite interesting and based on new evidence the authors have revised their original thesis regarding the density dependence. The analysis that makes use of the Keener-Tyson approximation (the ref should be cited on p.32 for eq 2) to estimate the parameter evolution in time is certainly novel and has strengthened the quantitative analysis substantially. It may also be possible to do a similar analysis with dark-field optics but is certainly far more difficult. The authors may like to more emphasize that precise tracing of the wavefront takes advantage of their high-resolution imaging. The new analysis that shows the timing difference in the density change and rotation speed change (Figure 6B) is quite compelling. Please indicate in the figure caption what the colors mean in panels Figure 6B and 6C. I assumed that they are the same as Figure 4 and 5.

Authors: "Motivated by this comment, we adopted the Martiel-Goldbeter model for exploration of our revised model. The data are shown in the new Figure 8. The Van Oss model was a motivating factor for the ABM model, and this has been cited in the revised document."

Reviewer response – I wonder whether a model based on Fitzhugh-Nagumo was even necessary to test Hypothesis 2. The Fitzhugh-Nagumo-based framework assumes cell-intrinsic excitability which is inconsistent with the current knowledge of the system and it would invite the need to modify the model with even more unnatural assumptions. Would the van oss model also do the job to test hypothesis 2? If so expected, it is better mentioned to indicate the degree of arbitrariness in the model selection. The extracellular cAMP-receptor loop and its degradation is central to the excitability of this system. The fact that the Martiel-Goldbeter-based model showed consistent results when compared to the experiments adds another confirmation of the thesis. Also please rewrite the last sentence of Figure 8 supplementary 1 legend. The waves and the feedback are cyclic but how they are described doesn't have to be so. The new experiments using agar block to pin the spiral core (Figure 9) are ingenious. Together with the new analysis in Figure 5D and Figure 8, I believe the revised manuscript has deciphered the nature of the excitability field as accurately as possible based on the current data. The implication that "cell state" is uniform across the field hints at what is the dominant factor that determines excitability – this may be the mean basal concentration of extracellular cAMP. As the wave period decreases there is less time for PDE to degrade extracellular cAMP. I would have liked to see a discussion along with the model interpretation but this may exceed the volume of the manuscript and thus could be put to test in future studies.

Authors: "Thank you for these comments. In response, we have gone into more depth about single-cell statistics. These are described, in-depth in Figure 1 – Supplementary Figure 3. In summary, we found no stable intrinsic variance between cells, but a slight history-dependent bias in the timing of cell firing between waves.

Regarding spiral wave formation. The implementation of the bPAC optogenetic tool gave some new insights into the formation of spiral waves. Specifically, cells that activate at the border of a region where cells are refractory and resting gave rise to broken waves that lead to spiral waves. These data and an illustrative schematic are shown in Figure 1- video 3 and Figure 1 – Supplementary Figure 2."

Reviewer response – The demonstration of spiral induction is very nice. However, I hope the authors carefully describe how this may relate to spirals that naturally arise in the cell population. Bias in the timing of the firing, biphasic response, and/or variability in the refractory period can play into this. These threads of idea and demonstration are not woven together in the Discussion section.

---

## [Author Response]

[Editors’ note: the authors resubmitted a revised version of the paper for consideration. What follows is the authors’ response to the first round of review.]

Reviewer #1 (Recommendations for the authors):[…]1. I would like to see more discussion about the criticism of what is presented as a key hypothesis in Höfer et al. (1995).I quote: "A previous mathematical model (Hofer, Sherratt and Maini, 1995), using the Martiel-Goldbeter model of cAMP release (Martiel and Goldbeter, 1987), suggested that this contraction is due to an increase in excitability due to an increase in cell density around the ring."I looked into the original paper, and I found that the local increase in excitability is due to the density-dependence of the cAMP dynamics (Eq. (4) in Höfer et al., in particular, the production rate \λ.phi(n)). In the model presented here, there is also a density-dependence in the cAMP production rate (Line 628), as each cell body acts as a source term for cAMP (when activated). So, I do not really see the conceptual difference between the model here and the one in Höfer et al. in terms of cAMP dynamics. Please elaborate.Ideally, the model would enable disentangling these effects (potential increase in excitability, and spatial reorganization) towards a stronger conclusion.In fact, quoting "However, spiral tip circulation holds a fixed radius in simulations without cell motion, suggesting that the change in cell spatial distribution is required to explain the contraction in tip circulation", it seems to me that the same explanation holds in Höfer et al., where the local increase in density gradually increases the wave frequency by sustained excitability.Further in the manuscript, quoting "We deduce that it is the rearrangement of cells via chemotaxis, not the increase in cell excitability, that drives vortex contraction." That is precisely my point: as far as I understand, in Höfer et al. the rearrangement induces higher excitability, and formally I noticed similarities in the modeling of cAMP dynamics.A related observation follows from the quotation "We note that other, more complicated dynamics of external cAMP have been used (e.g. active degradation by cells Hofer, Sherratt and Maini, 1995), however here we chose to omit such potential effects in the interest of simplicity."I think it is worth checking that it does not change the dynamics, even if cells actively degrade/consume cAMP.

The new submission now directly addresses the Hofer model and its limitations using experimental data. This is documented in the revised manuscript in a section entitled “Hypothesis 1” with the data arguing against this model displayed in Figure 6.

Regarding the modelling assumptions, the excitability does not explicitly change. This is now described in the main text under “Hypothesis 2”. For the model selection- this extends existing models, but incorporates a newer model of cAMP release, as now outlined in the main text under “Hypothesis 2”. The Supplementary Information outlines the selection process for the model of cell motion. The Supplement also now includes a description of the inertia effect, and how this allows cells to essentially “ignore” the back of the wave.

2. The chemotaxis models differ significantly from classical (basic) models, for it includes inertia. I understand that the authors follow a data-driven approach, extending the standard model to better match the observed trajectories in a sampled area. Then, it is very satisfactory to reproduce the full macroscopic picture.I suspect that inertia plays an important role, as it encodes memory effects in the cell response to the signaling gradients. Would it be possible to emphasize the important features of the chemotactic model? Practically, would it be possible to link some of the chemotactic parameters to the frequency threshold beyond which the cells follow streams (Figure 2)?Similarly, it would be highly valuable, I think, to compare the results in Figure 4D with the model.

The revised manuscript has moved past this model, based on optogenetic and mechanical perturbation experiments that favour a new model more consistent with the data. The chemotaxis features of the model have been articulated more fully in the SI.

Regarding frequency thresholds, we did not select for models in the context of the periodic waves distant from the spiral core. There is a detailed mathematical model that can quantitatively reproduce the motion of cells in a wide range of signalling contexts (PMID: 25249632), but our motivation here was to adopt the simplest model that was consistent with our spiral core data.

Reviewer #2 (Recommendations for the authors):[…]This was a very enjoyable piece of work and I congratulate the authors on this achievement. Scientifically, however, I see several major caveats that I hope the authors can address when revising the manuscript.Figure 1 shows imaging results of the onset of collective signaling. This is a great refinement of the work by Gregor et al. (Science 2010) where they have looked at small populations of cAMP FRET sensor expressing cells and analyzed firing events and different spatial locations. Whether random firing occurs prior to synchronous firing was not clear in Gregor et al. but was inferred from stochastic firing response when cells were stimulated at sub-nanomolar extracellular cAMP using a perfusion chamber. The present work provides much stronger evidence of this sequence of events proposed by Gregor et al. where the accumulation of stochastic firing at the single-cell level leads to synchronous firing. The FRET measurements by Gregor lacked the necessary S/N ratio and such fine resolution dynamics at the population level at the time were hard to come by. The authors' use of Flamindo is thus a very nice approach. However, again authors failed to mention that the use of Flamindo to analyze wave dynamics is not without precedent (Hashimura et al., Communications Biology 2, 34, 2019). This paper should be cited.

Many thanks for the positive appreciation of the experimental approach. The Hashimura paper has now been referenced when we introduce Flamindo2.

Figure 2 addresses the link between the wave and cell migration. Here and elsewhere in the text, I am afraid that the present manuscript uses the term 'information' quite loosely. It is worth noting that the information processing property of this system has been analyzed quite extensively, and this could have been elaborated better in the text with a more balanced view of various possibilities. The current observations should be discussed in light of these single-cell level input-output relations. The fact that frequency increases at the timing of streaming and the timing of spiral evolution has been shown by Antony Durston ("The control of morphogenesis in *Dictyostelium* discoideum" in Eucaryotic Microbes as Model Developmental systems 1977) and more accurately by others as described in the public review. Some novelty I see in Figure 2 data is the quantitative analysis of the spiral core size and change in the frequency. However, from simple observations, as shown in the present paper, it is not possible to decipher causality from correlated events. The caveat is that one cannot discriminate the effect of the number of waves experienced by the cell versus the frequency, periodicity versus wave speed. In the text, the former is completely ignored or maybe they are mixed up. I don't see clear evidence of frequency modulation. Early works to tackle this challenge were pioneered by Michael Vicker and then by David Soll and his colleagues that showed with the use of perfusion setup that cells read out temporal information rather than the spatial gradient. This aspect was further vindicated by Nakajima et al. (Nat. Commun. 5, 5367, 2014) that employed an isolated single-passage of artificially generated wave that showed that it is the wave passage time (wave speed) that dictates what is being read out. The current understanding derived from this study and Skoge et al. (PNAS ) is that the adaptive response by the Local Excitation Global Inhibition (LEGI) framework (Levchenko & Iglesias) is fully capable of describing the readout property. At a slow wave speed, cells are provided enough time to readout spatial gradient at the steady-state response. At fast waves, the directionality is first determined by the first hit mechanism which is based on the delay of stimulus arrival and the following transduction across the cell. This makes use of transient response and requires temporally increasing extracellular cAMP thus occurring specifically at the wavefront and not waveback. Much faster wave speed effectively makes the stimulus spatially uniform not allowing cells to form a localized leading edge. What does require an extension to the LEGI framework is this fast-wave regime. At the 3-min interval stimulus, the system comes somewhat close to the limit capable to make use of the first hit mechanism. One possible resolution to this paradox is the hysteresis/memory effect. The authors cite Skoge et al., which is very important here for data interpretation, but there is another work as relevant that employed artificial rotating waves to induce circular cell migration (Nakajima et al., Lab Chip 22, 4382, 2016; Figure 6f). There, 4-5 passages of 5.5-min periodicity waves can induce cell polarity or the memory effect where cells become elongated and do not stop at the wave back. Given this prior observation, I just don't see the need for a 3-min frequency-specific collective migratory response as the authors seem to suggest. Given intermediate wave speed stimulus first, cells will obtain this highly polarized state and thus could move by "steering" mode of chemotaxis around the spiral core. It could be that 3.5 min periodicity is a consequence of cells taking high polarity – in this case, the causality is reversed. I believe the manuscript would be greatly improved if suggest authors take these possible scenarios into consideration in the Discussion section.

Thanks for pointing out these issues. Importantly, we have now carried out a substantial series of experimental perturbations of wave frequency, using optogenetic activation of the bPAC adenylyl cyclase. Specifically, we generated trains of pulses of different fixed frequencies and also, trains of gradually increasing frequency. The data (presented in the new Figure 3) argue strongly for migration specified by the frequency by which waves are generated by the signalling centre. Additional data corroborating these effects in the unperturbed state are shown in Figure 6B. In particular, we also show that many facets of the wave properties (width, speed) are determined by the frequency at which waves are generated from the signalling centre (Figure 5B). As the Reviewer suggested, we have generalised these new results in the discussion, in the context of the diverse literature the Reviewer highlights.

Figure 3. The fact that cells move towards the incoming spiral waves is well studied and established. I am not sure what the authors wanted to test here (L. 206-209).

This was a figure to highlight the methods we were applying. It has now been merged with key data on spiral progression, in the new Figure 4.

Figure 4 addresses the curvature and the dispersion relation. The quantification of the dispersion relation at this resolution is certainly nice, however, the authors failed to mention that the curvature relation has been characterized earlier by Foerster Muller Hess "Curvature and spiral geometry … (Development 109, 11-16, Figure 6 for the spiral geometry fitting and Figure 3,4 for the dispersion curve). This paper is cited in this manuscript but only in general terms and not this specific point which was essentially the central theme of the Foerster paper. I could well be missing something, but since the authors have not put the results in the light of the earlier finding, the importance of the present data is hard to grasp. In the figure legend, it refers to ref. [41] for the parameter epsilon, however, the reference is not numbered.

This relates to a subsection of what is now Figure 5. The importance of the analysis is that we can measure the fundamental signalling parameters (eg. excitability, planar wave speed) and how their changes drive spiral wave progression. We have cited the Foerster paper in this section- this study showed that a cAMP spiral that does not change structure and dynamics can satisfy the eikonal relation. However, our high-resolution dataset allowed use this approach to attain signalling parameters throughout the progression of the spiral, at high time resolution. Therefore we can track how the spiral changes over developmental timescales, and how the fundamental signalling parameters change to drive developmental progression.

Figure 5 and L. 300-306 describe model simulations and the cell density effect. By comparing model simulations with and without cell motion (Figure 5D and 5E), the authors suggest that the change in cell distribution is required to explain the contraction in tip circulation. They suggest a scheme where the cellular ring constrains spiral tip motion and the extracellular cAMP field guides cells more inwards. How this scheme differs from those proposed by others (Hoefer and Maini) is not straightforward. This is hard to test experimentally and may be ill-posed. Making cells immobile by F-actin destabilizing drug and substrate adhesion also affects the cAMP synthesis (Interface 13 20160233, 2016), and thus cell motility and the cAMP-relay is highly intertwined and the parameters are mutually dependent. Cells treated with caffeine which inhibits cAMP synthesis in *Dictyostelium* do show open rings that fail to shrink and the core size depends on caffeine concentration (Siegert and Weijer JCS 93, 325-335, 1989). There, the cells are moving but the ring fails to contract and thus contradicts the authors' present thesis. What is missing here is an experimental demonstration of the side effects of caffeine that affect chemotaxis and cell motility. It would be very interesting to apply the current analysis to caffeine-treated cells. For example, caffeine affects the TORC2 pathway. But again, this is very hard to test as both chemotaxis and the cAMP-relay share a common signaling pathway.

In the revised manuscript, we have been explicit about the distinctions and similarities between the Hofer model and the cell patterning model. Each model has its own subsection in a section entitled “Hypotheses”. We have also proposed an additional hypothesis, which contrasts both models.

Regarding, experimental testing, we have incorporated new data where we explicitly evaluated the Hofer model (cell density). These data are shown in Figure 6. Specifically, we see no strong coupling between cell density and either excitability or wave period. Spiral wave progression proceeds (and is largely completed) prior to changes in density.

To test the ring closure hypothesis, while avoiding the caveats of drug treatments, we devised a mechanical perturbation experiment, in which we block the hole closure using and agarose block (Figure 9). This experimental test revised our thinking of the mechanism. We strongly thank the reviewer for encouraging us to think about this.

Sgro 2015 and Kamino 2017 (L. 304) do examine cell-density independence of oscillatory instability arising from fold-change response characteristics. It addresses important but rather idealized general conditions necessary for robustness to cell density fluctuations and thus it may not be adequate for the more system-specific detailed analysis required here. The abstract nature stems from the fact that density dependence comes mainly at the level of cAMP detection and synthesis. In reality, how extracellular PDE production is regulated most likely changes this cell density dependence drastically, and thus the use of these models here does not warrant a meaningful conjecture. cAMP wave geometries do depend strongly on cell density as has been demonstrated by Lee, Cox, and Goldstein (Phys Rev Lett 76, 1996). The Martriel-Goldbeter model is not just a model of cAMP release (L.301). It describes cAMP synthesis, cAMP degradation, and receptor adaptation. It is still the most consistent model with the biochemistry bulk-based analysis by Peter Devreotes and single-cell-based cAMP-relay analysis as described below. The direct comparison of the current simulations should also be made with the partial differential equation-agent hybrid-based simulations by van Oss et al. Weijer (J theor Biol 181, 203-213, 1996) which are very similar in approach and methodology except that it is based on Martiel-Goldbeter model. This important work is not discussed nor cited. They have made a good analysis of the relationship between the wave speed, cell movement, and cell density around the spiral core (Figure 3 of this reference). Wave speed increases with increasing cell density and the core shrinks.

Motivated by this comment, we adopted the Martiel-Goldbeter model for exploration of our revised model. The data are shown in the new Figure 8. The Van Oss model was a motivating factor for the ABM model, and this has been cited in the revised document.

Figure 7 addresses cell-cell variability during spiral development. The collective cAMP oscillations/waves in *Dictyostelium* require a receptor-mediated extracellular feedback loop both for amplification (via cAMP) and also for de-adaptation (clearing of extracellular cAMP by extracellular PDE). Dependence on the cAMP receptors is well-addressed by various hybrid receptors and mutations (Dormann et al. JCS 114, 2513-, 2001). Temperature shift mutation of ACA can turn waves on and off (Patel et al. EMBO 19, 2247-,2000). Oscillations can also be turned on or off in the PDE-null mutant by mimicking degradation by assisted dilution by perfusion (Masaki et al., Biophys 104, 1191-, 2013). The major unresolved question regarding the system is how far we understand these properties at the single-cell level. There are some cell-cell statistics in this work but they can be highlighted. This is important as how waves are initiated and how spirals develop do depend heavily on cell-cell variability in the related parameters along the developmental path (Lauzeral & Goldbeter PNAS, Levine et al. PNAS 1996, Sawai, Thomason & Cox Nature 2005). At the single-cell level, the average response characteristics are well characterized (Gregor et al. Science 2010, Kamino et al. PNAS). The response dynamic as well as the fold-change conditions necessary for the amplification is consistent with the macroscopic behavior of the system. As for the refractory period, dependence on ligand clearing (which is what PDE does) has been quantified at the single-cell level (Fukujin et al., Interface 13 20160233, 2016). Here it is shown that 10 min of ligand clearing is required for full response recovery while at 3.5 min washing, the response is attenuated to about 50% in amplitude. The absence of the absolute refractory period and the fact that cells can still respond after 3.5 min ligand clearing should be noted and discussed (L. 354- and elsewhere). Also, according to Gregor et al. repetitive stimulus at 3 min intervals will eventually attenuate the cAMP-relay response. Could this be related to the final extinction of the waves? All these important points are highly relevant when the analysis is zoomed in at the single-cell level resolution.

Thank you for these comments. In response, we have gone into more depth about the single cell statistics. These are described, in depth in Figure 1 – Supplementary Figure 3. In summary, we found no stable intrinsic variance between cells, but a slight history dependent bias in the timing of cell firing between waves.

Regarding spiral wave formation. The implementation of the bPAC optogenetic tool gave some new insights into the formation for the spiral waves. Specifically, cells that activate at the border of a region where cells are refractory and resting gave rise to broken waves that lead to spiral waves. These data, and an illustrative schematic are shown in Figure 1- video 3 and Figure 1 – Supplementary Figure 2.

[Editors’ note: what follows is the authors’ response to the second round of review.]

The manuscript has been improved but there are some remaining issues that need to be addressed, as outlined below:The reviewers have discussed their reviews with one another, and the Reviewing Editor has drafted this to help you prepare a revised submission. While both reviewers appreciate the improvement and I depth modifications made to this version of the manuscript they still raise significant concerns that should be addressed prior to publication.Clarifications must be added to explain the modeling:The authors draw three hypotheses:1. A "global" scenario, where the overall signaling dynamics are coupled with cell density, essentially following Hofer et al. Thus, cell reorganization through chemotaxis can drive changes in the excitability properties of the (inhomogeneous) medium, with a significant increase at the center when cells aggregate. This scenario is ruled out based on a predicted temporality (cell massive reorganization precedes changes in wave dynamics) which is in contrast with the experimental observations.2. A "local" scenario, where the local reorganization of cells near the tip dramatically reduces both its radius and its speed (but not its rotation period, which is a drawback). The conceptual difference with the model of scenario 1 is not obvious. The models are of a different nature (continuous field versus the individual description of the cell population). Nevertheless, the release of external cAMP (bottom of p.33) increases with the local cell density, which could be put in similarity to the work by Hofer et al. In the latter model, the cell density-dependent "excitability" is motivated as follows: "An increase in cell density increases the local production and degradation rates per volume element, and it decreases locally the share of the extracellular volume in a volume element."Thus, it is not justified as an increase of "individual excitability", as the authors describe it here on p.14: "In this view, the cell excitability increases by virtue of cells accumulating at the spiral core through chemotaxis". In fact, in Hofer et al., the increase in cell density changes the excitability locally in the medium, but this might not be interpreted as an increase in individual cell excitability (in contrast with Hypothesis #3).

This has been corrected, see below.

3. A "molecular-only scenario", with no coupling with the cell density. The idea is that the cells modify their individual excitability state when waves are passing through. The additional equation taking into account this new idea is minimal, which is nice. The cell re-organization follows from chemotaxis, but there is no feedback in the model (actually, the cell density is not even described, see Figure 8).Comments (towards a clarification of the three hypotheses):Overall, the modelling part is convincing. It could be interesting to present the two alternatives (providing that Hyp #1 and Hyp #2 are not conceptually different, see below) as follows:"emerging collective excitability by cell reorganization with constant individual excitability" (#1 and #2) versus "adaptative individual excitability in a homogeneous cell density".Is that (nearly) correct?

In agreement with this suggestion, the original Hypotheses 1 and 2 have been grouped into Hypothesis 1 (Hypothesis 3 is now Hypothesis 2). “Hypothesis 1: increased collective excitability by cell reorganisation with constant individual excitability.” (Line 283) and “Hypothesis 2: increased individual cell excitability driven by wave dynamics.” (Line 343).

The agent-based model of Hypothesis #2 may not be equivalent to the Hofer et al. model, but what are the exact conceptual differences between them? Along this line, the results of Hypothesis #2 may have similar inconsistencies with the observations in Hypothesis #1. Indeed, it looks like in Figure 7, the cell density in the model shows dramatic changes after 1h. The authors argued this was a clear way to remove Hypothesis #1. Isn't it the case here also?This comment is of little importance for the conclusion of the paper, as both Hypotheses #1 and #2 are ruled out after all. But this is a source of confusion.

Beyond the different models of cAMP release, there is little conceptual difference between our ABM model and the model of van Oss and of Hofer. The grouping of these hypotheses resolves this cause of confusion.

Another source of confusion was the apparent arbitrariness in the choice of the molecular "excitable" systems in the two hypotheses 2 and 3. The Martiel-Goldbeter (MG) system is used to test hypothesis 2, whereas, the more basic Fitzhugh-Nagumo (FHN) system is used to test Hypothesis 3. This may be not an important drawback, since the two models belong to the same class of excitable systems. However, this may add some confusion and some possible weakness to the overall conclusions.It is understood that the authors followed a recent modelling approach for testing hypothesis 2 (although the cAMP internal dynamics are somewhat basic equations of FHN type, as compared to the MG model). However, they used some modified version of an older series of work from the late 1980s to test hypothesis 3.

Our model selection process was not arbitrary – however our manuscript did not reflect our line of logic. We have now adjusted the narrative to explain our reasoning (Lines 304-309 and 365-374).

To expand here: the models analysed by Hofer et al. (PDE) and van Oss et al. (ABM) extended the model analysed by Tyson et al. (1989) by considering changes in cell density. Being the first (to our knowledge) to analyse the spatial pattern of cAMP waves across homogeneous density of cells, the Tyson model can be considered to be the precursor for the Hofer and van Oss models.

As the models of Hofer and van Oss (based on the Martiel-Goldbeter model for cAMP release) are inconsistent with our experimental data, the motivation of our ABM was to attempt to reconcile our data with the hypothesis that cells do not change their intrinsic excitability. To this end, we adopted a promising alternative model for cAMP release, the modified FHN model that Sgro extensively tested experimentally. Like the Hofer and van Oss model, this approach failed to reproduce our experiments. We included this analysis in the paper for completeness.

We then went back to the original model by Tyson (no density fluctuations) and then found that our experimental observations could be reproduced in the simplest way by allowing the excitability to depend on the external [cAMP] – motivated by our experiments. In summary, extending the Tyson model with adaptive cell density (van Oss, Hofer and our ABM) is inconsistent with our data, however extending the Tyson model with adaptive cell excitability (our PDE) is consistent with our data.

Moreover, the formulation of the model in hypothesis 2 is ambiguous:"We adopt the model by Tyson et al. (Tyson and Murray, 1989), which is of the general form of continuum excitable media (equation (1)) with the dynamics of cAMP release (functions f and g in equation (1)) governed by the Martiel-Goldbeter model…"However, there is nothing like "f and g" in equation (1), and moreover, the MG system (6) is a closed one, and equation (1) should be viewed as an asymptotic limit capturing the geometric features of the system (6).

These were referencing an equation from a previous draft and have now been removed. (Line 804).

Also, references are missing in the section "General theory of excitable media" on page 32. I understood that the main reference is Tyson-Keener (1988), but the equation (1) as written in the present ms, is seemingly written as N = N0 + DK [Tyson-Keener 1988, page 10, left column]. The additional parameter epsilon in (1) is dimensionless, so it would make perfect sense if supported by a suitable reference (and it has a clear biological interpretation).

The relevant references (Tyson and Keener 1988, Keener 1986 and Winfree 2001) have now been included (section starting at line 695).

The authors also don't describe how this excitability change comes about – likely the level of extracellular cAMP. The oscillations and waves in this system may be taking place at the nanomolar range at the basal or several hundred nanomolar ranges at the basal as long as there is enough fold-change increase. One would assume that higher cell density is necessary to support a high basal level state however if the period decreases degradation of cAMP fails to catch up with the frequency. This cannot be read from the intracellular cAMP indicator and is important to interpret from the model.

There is a paragraph in the discussion relating to gene expression changes in response to cAMP. We have extended this to include the reviewers suggestion about cAMP increasing without gene expression changes (Lines 544-546).

Reviewer #1 (Recommendations for the authors):I consider this manuscript as a new submission since the conclusions have been significantly changed. I found the new version highly valuable, with a new hypothesis that is conceptually different than the previous ones and explains most of the observations. Moreover, this new hypothesis has been tested with an original experiment, and it succeeded where the previous paradigm failed.Main comment:The authors draw three hypotheses:1. A "global" scenario, where the overall signalling dynamics are coupled with cell density. Thus, cell reorganization through chemotaxis can drive changes in the excitability properties of the (inhomogeneous) medium, with a significant increase at the centre when cells aggregate.This scenario is ruled out based on a predicted temporality (cell massive reorganization precedes changes in wave dynamics) which is in contrast with the experimental observations.2. A "local" scenario, where the local reorganization of cells near the tip dramatically reduces both its radius and its speed (but not its rotation period, which is a drawback). I'm afraid I don't quite see the conceptual difference with the model of scenario 1. The models are of a different nature (continuous field versus the individual description of the cell population). Nevertheless, the release of external cAMP (bottom of p.33) increases with the local cell density, which could be put in similarity to the work by Hofer et al. In the latter model, the cell density-dependent "excitability" is motivated as follows: "An increase in cell density increases the local production and degradation rates per volume element, and it decreases locally the share of the extracellular volume in a volume element."Thus, it is not justified as an increase of "individual excitability", as the authors describe it here on p.14: "In this view, the cell excitability increases by virtue of cells accumulating at the spiral core through chemotaxis".I would put it like this: in Hofer et al., the increase in cell density changes the excitability locally in the medium, but this might not be interpreted as an increase in individual cell excitability (in contrast with Hypothesis #3).I don't claim that the agent-based model of Hypothesis #2 is equivalent to the Hofer et al. model, but I don't see conceptual differences between them. In this line, I was asking myself if the results of Hypothesis #2 do not suffer the same inconsistency with observations as of Hypothesis #1. Indeed, it looks to me that in Figure 7, the cell density in the model shows dramatic changes after 1h. The authors argued this was a clear way to remove Hypothesis #1. Isn't it the case here also?This comment is of little importance for the conclusion of the paper, as both Hypotheses #1 and #2 are ruled out after all. But this was a source of confusion for me.3. A "molecular-only scenario", with no coupling with the cell density. The idea is that the cells modify their individual excitability state when waves are passing through. The additional equation taking into account this new idea is minimal, I appreciate it. The cell re-organization follows by chemotaxis, but there is no feedback in the model (actually, the cell density is not even described, see Figure 8).I found the modelling part very convincing. I may suggest presenting the two alternatives (providing that Hyp #1 and Hyp #2 are not conceptually different, but this is my own understanding) as follows:"emerging collective excitability by cell reorganization with constant individual excitability" (#1 and #2) versus "adaptative individual excitability in a homogeneous cell density".Is that (nearly) correct?

These comments have been addressed above, by combining Hypotheses 1 and 2, as suggested by the reviewer.

"Altogether, our results describe how the collective behaviour of 10^5 -10^6 cells across several millimetres is organised by the signalling dynamics of a few hundred cells."– I'm not sure I understand what is meant here. I agree that the whole organization relies on a "pacemaker", that may be constituted of a few hundred cells (although I don't know what supports this number, is there a spatial threshold to decide where is the pacemaker?). In particular, as far as I understand the model, the cells are all equally capable of producing and relaying the signal in the same terms. It is thus the geometry of the process, and not the cell heterogeneity (leaders vs. followers) which drive the dynamics.I would like to make it clear that "the few hundred cells" are not considered as different from the other in terms of the capability of making the pattern (would there be some artificial exchanges within the population)? But maybe I got it wrong.

We have reworded this sentence to avoid the possible interpretation that there are inherent differences between cells. The new sentence is “Altogether, our results describe how the collective behaviour of 10^5^-10^6^ cells across several millimetres is organised by circulation of a self-sustaining signal over a 10^2^-10^3^ cells.” (Line 498).

Figure 7/8 (titles) – They are both mathematical models, so the titles should be updated (in reference to the corresponding hypotheses 2/3 maybe?).

The titles have been changed to ”Figure 7: Mathematical model of spiral wave progression (constant cell excitability, variable cell density)” and “Figure 8: Mathematical model of spiral wave progression (constant cell density, variable cell excitability)”.

Figure 4-D-1st and Figure 6-B-2nd – I understand that this is the same figure. Is that correct? (a few points are different, though).

We have added a line to the legend for Figure 6B to explain how these data are related: “Shown are the same 3 datasets (colours) shown in Figure 4 and 5. Data points represent average values per rotation, in contrast to Figure 4D.” We have added text to the legend of 4D to be more explicit about the data presented “Time series of the spiral tip circulation rotational period calculated the instantaneous rotation”.

In the methods, it seems that the models which are used in the simulations are presented in the reversed order (Hypothesis #3, then Hypothesis #2). Is there a good reason for this?

We have changed the order of these, in response to this query.

Reviewer #2 (Recommendations for the authors):Authors: "Thanks for pointing out these issues. Importantly, we have now carried out a substantial series of experimental perturbations of wave frequency, using optogenetic activation of the bPAC adenylyl cyclase. Specifically, we generated trains of pulses of different fixed frequencies and also, trains of gradually increasing frequency. The data (presented in the new Figure 3) argue strongly for migration specified by the frequency by which waves are generated by the signalling centre. Additional data corroborating these effects in the unperturbed state are shown in Figure 6B. In particular, we also show that many facets of the wave properties (width, speed) are determined by the frequency at which waves are generated from the signalling centre (Figure 5B). As the Reviewer suggested, we have generalised these new results in the discussion, in the context of the diverse literature the Reviewer highlights. "Reviewer response – The authors have shown that 3 min periodic forcing gives rise to conglomerates of many aggregates which they interpret as backing their thesis that fast frequency supports the collective motion. The other likely possibility is that 3 min periodic forcing entrains other wave territories and this demonstration while it is very nice does not prove by itself that fast frequency is necessary for the transition to collective motion.

We have added a sentence including this interpretation: “These artificial signalling centres induce large aggregates by entraining the territories of neighbouring signalling centres.” (lines 210-212).

Authors: "Hofer model and the cell patterning model. Each model has its own subsection in a section entitled "Hypotheses". We have also proposed an additional hypothesis, that contrasts both models.Regarding, experimental testing, we have incorporated new data where we explicitly evaluated the Hofer model (cell density). These data are shown in Figure 6. Specifically, we see no strong coupling between cell density and either excitability or wave period. Spiral wave progression proceeds (and is largely completed) prior to changes in density.To test the ring closure hypothesis, while avoiding the caveats of drug treatments, we devised a mechanical perturbation experiment, in which we block the hole closure using an agarose block (Figure 9). This experimental test revised our thinking of the mechanism. We strongly thank the reviewer for encouraging us to think about this."Reviewer response – The new data is quite interesting and based on new evidence the authors have revised their original thesis regarding the density dependence. The analysis that makes use of the Keener-Tyson approximation (the ref should be cited on p.32 for eq 2) to estimate the parameter evolution in time is certainly novel and has strengthened the quantitative analysis substantially. It may also be possible to do a similar analysis with dark-field optics but is certainly far more difficult. The authors may like to more emphasize that precise tracing of the wavefront takes advantage of their high-resolution imaging. The new analysis that shows the timing difference in the density change and rotation speed change (Figure 6B) is quite compelling. Please indicate in the figure caption what the colors mean in panels Figure 6B and 6C. I assumed that they are the same as Figure 4 and 5.

This has been carried out. The text “Shown are the same 3 datasets (colours) shown in Figure 4 and 5. Data points represent average values per rotation, in contrast to Figure 4D”. in now included in the legend to Figure 6.

Authors: "Motivated by this comment, we adopted the Martiel-Goldbeter model for exploration of our revised model. The data are shown in the new Figure 8. The Van Oss model was a motivating factor for the ABM model, and this has been cited in the revised document."Reviewer response – I wonder whether a model based on Fitzhugh-Nagumo was even necessary to test Hypothesis 2. The Fitzhugh-Nagumo-based framework assumes cell-intrinsic excitability which is inconsistent with the current knowledge of the system and it would invite the need to modify the model with even more unnatural assumptions. Would the van oss model also do the job to test hypothesis 2? If so expected, it is better mentioned to indicate the degree of arbitrariness in the model selection. The extracellular cAMP-receptor loop and its degradation is central to the excitability of this system. The fact that the Martiel-Goldbeter-based model showed consistent results when compared to the experiments adds another confirmation of the thesis.

As mentioned above, we used the FHN model in an attempt to reconcile models of constant cell excitability with our data.

Also please rewrite the last sentence of Figure 8 supplementary 1 legend.

This has been rewritten.

The waves and the feedback are cyclic but how they are described doesn't have to be so. The new experiments using agar block to pin the spiral core (Figure 9) are ingenious. Together with the new analysis in Figure 5D and Figure 8, I believe the revised manuscript has deciphered the nature of the excitability field as accurately as possible based on the current data. The implication that "cell state" is uniform across the field hints at what is the dominant factor that determines excitability – this may be the mean basal concentration of extracellular cAMP. As the wave period decreases there is less time for PDE to degrade extracellular cAMP. I would have liked to see a discussion along with the model interpretation but this may exceed the volume of the manuscript and thus could be put to test in future studies.

We have added a short section to introduce these ideas in the discussion. “A change in cell excitability could also arise independent of changes in gene expression. For example we can imagine a scenario in which the degradation of cAMP cannot keep up with the increased wave frequency. In this situation, the baseline cAMP increases over time.” (Lines 544-546).

Authors: "Thank you for these comments. In response, we have gone into more depth about single-cell statistics. These are described, in-depth in Figure 1 – Supplementary Figure 3. In summary, we found no stable intrinsic variance between cells, but a slight history-dependent bias in the timing of cell firing between waves.Regarding spiral wave formation. The implementation of the bPAC optogenetic tool gave some new insights into the formation of spiral waves. Specifically, cells that activate at the border of a region where cells are refractory and resting gave rise to broken waves that lead to spiral waves. These data and an illustrative schematic are shown in Figure 1- video 3 and Figure 1 – Supplementary Figure 2."Reviewer response – The demonstration of spiral induction is very nice. However, I hope the authors carefully describe how this may relate to spirals that naturally arise in the cell population. Bias in the timing of the firing, biphasic response, and/or variability in the refractory period can play into this. These threads of idea and demonstration are not woven together in the Discussion section.

In response to this suggestion, we have added a sentence summarising these potential causes of natural spiral wave formation “This experiment suggests that spiral waves might naturally arise from spontaneous cell activation behind a circular wave, where neighbouring cells are transitioning from a refractory to rest state.” (lines 147-149).